# The impact of preceding convection on the development of Medicane Ianos and the sensitivity to sea surface temperature

Claudio Sánchez[1], Suzanne Gray[2], Ambrogio Volonté[2,3], Florian Pantillon[4], Ségolène Berthou[1], and Silvio Davolio[5,6]

[1]MetOffice, Exeter, United Kingdom
[2]Department of Meteorology, University of Reading, UK
[3]National Centre for Atmospheric Science, University of Reading, Reading, UK
[4]LAERO, Université de Toulouse, CNRS, UPS, IRD, Toulouse, France
[5]Dipartimento di Scienze della Terra, Università degli Studi di Milano, Milan, Italy
[6]Institute of Atmospheric Sciences and Climate, National Research Council, CNR-ISAC, Bologna, Italy

**Correspondence:** Claudio Sánchez (claudio.sanchez@metoffice.gov.uk)

**Abstract.** Medicane Ianos in September 2020 was one of the strongest medicanes observed in the last 25 years. It was, like other medicanes, a very intense cyclone evolving from a baroclinic mid-latitude low into a tropical-like cyclone with an axisymmetric warm core. The dynamical elements necessary to improve the predictability of Ianos are explored with the use of simulations with the Met Office Unified Model (MetUM) at 2.2 km grid spacing for five different initialisation times, from four to two days before Ianos's landfall. Simulations are also performed with the Sea Surface Temperature (SST) uniformly increased and decreased by 2 K from analysis to explore the impact of enhanced and reduced sea-surface surface fluxes on Ianos's evolution. All the simulations with +2 K SST are able to simulate medicane Ianos, albeit too intensely. The simulations with control SST initialised at the two earliest times fail to capture intense preceding precipitation events at the right locations, and the subsequent development of Ianos. Amongst the simulations with -2 K SST, only the one initialised at the latest time develops the medicane.

Links between sea-surface fluxes and upper-level baroclinic processes are investigated. We find three elements that are important for Ianos's development. First, an area of low-valued potential vorticity (PV), termed a "low-PV bubble", formed within a trough above where Ianos developed; diabatic heating associated with a preceding precipitation event triggered a balanced divergent flow in the upper-levels which contributed to the creation and maintenance of this low-PV bubble as shown by results from a semi-geostrophic inversion tool. Second, a quasi-geostrophically ascent was forced by mid and upper levels during Ianos's cyclogenesis, it is partially associated with the geostrophic vorticity advection, which is enhanced by the growth and advection of the low-PV bubble. Third, diabatic heating dominated by deep convection formed a vertical PV tower during Ianos's intensification and continued to produce diabatically-induced divergent outflow aloft, thus sustaining Ianos's development. Simulations missing any of these three elements do not develop medicane Ianos.

Our results imply the novel finding that preceding convection was essential for the subsequent development of Ianos, highlighting the importance of the interactions between near-surface small-scale diabatic processes and the upper-level quasi-

geostrophic flow. A warmer SST strengthens the processes and thus enables Ianos to be predicted in simulations initiated at the earlier times.

# 1  Introduction

The Mediterranean region is one of the most cyclogenetic areas of the world (Hoskins and Hodges, 2002; Wernli and Schwierz, 2006). The most intense cyclones there produce considerable hazards, such as heavy precipitation (Flaounas et al., 2022; Khodayar et al., 2018), extreme winds (Raveh-Rubin and Wernli, 2015; Lfarh et al., 2023) or coastal impacts from large sea-level anomalies (Lionello et al., 2019; Ferrarin et al., 2023). The region is also a climate change hot-spot with large projected
changes in temperature and precipitation (Giorgi, 2006), which might alter the mechanisms of Mediterranean cyclones and their hazards.

Some of the most intense Mediterranean cyclones exhibit tropical-like characteristics such as a cloud-free eye and an axisymmetric warm core, and hence they are commonly termed "Mediterranean hurricanes" or "medicanes". Cavicchia et al. (2014) developed a climatology of this phenomenon from the dynamical downscaling of a high resolution reanalysis. They
concluded that medicanes occur one to three times per year, and their genesis is mostly in the western Mediterranean and in the region extending between the Ionian Sea and the North-African coast. Medicanes have a peculiar seasonal cycle, their peak is at the beginning of winter, a relevant number of events occur during autumn, a few over spring, and they have very low activity in summer. Hence, medicanes tend to occur when there is cold air aloft.

The upper-level cold air is provided by the intrusion of upper-level disturbances such as potential vorticity (PV) streamers
or cut-off lows. The cyclogenesis of Medicanes thus occur under such moderate to strong baroclinic environments (Mazza et al., 2017; Fita and Flaounas, 2018; Flaounas et al., 2022). For instance, the reduction of the PV streamer by a PV inversion technique suppressed the cyclogenesis of a medicane in the cases described in Homar et al. (2003) and Carrió et al. (2017). Latent heating due to convection is also an important player in the intensification of Medicanes. Fita and Flaounas (2018) quantified the development of the surface pressure tendency for a medicane in December 2005 and attributed its deepening
to warming in the atmospheric column from diabatic heating and advection within the upper troposphere. Convection is often fuelled by air-sea fluxes from the sea surface, e.g. the ensemble simulations of Noyelle et al. (2019), with different Sea Surface Temperature (SST) anomalies, relate the warm core strength to a linear increase of SST warming, but they also reveal that growth in the minimum pressure depth at maximum intensity is non-linear with the SST increase. Miglietta et al. (2011) found that increasing SST leads to a deeper medicane, stronger surface wind speeds and longer life-time of the medicane's tropical
features.

The relative importance of baroclinic instability and diabatic heating in medicane intensification seems to be case dependent. Some medicanes may not undergo a full tropical transition (Davis and Bosart, 2003), as suggested by Fita and Flaounas (2018)

where their case classified as a subtropical cyclone. The climatology of tropical transitions of McTaggart-Cowan et al. (2013) shows high frequency of occurrence of strong tropical transition over the Mediterranean Sea, a region with a moderate to high coupling index McTaggart-Cowan et al. (2015). Miglietta and Rotunno (2019) proposed a classification of medicanes into three categories: (A) those where their later stages are dominated by the Wind-Induced Surface Heat Exchange mechanism (WISHE Emanuel, 1986; Rotunno and Emanuel, 1987), (B) those where the baroclinic instability is also important at later stages, and (C) a blend of the previous two where tropical transition and a dramatic intensification occurs after a short but intense interaction of the cyclone with an upper-level PV streamer. Miglietta and Rotunno (2019) conclude that the presence of a symmetric deep warm core does not imply full tropical dynamics and hence the terms "tropical-like" transition or "Mediterranean tropical-like cyclone" are often employed in the medicane literature.

A reliable prediction of medicanes and their hazards can be obtained from a Numerical Weather Prediction (NWP) model able to reproduce the cyclone intensification mechanisms adequately and initialised from a realistic analysis. The predictability of seven medicanes was assessed in Di Muzio et al. (2019) with an operational ensemble NWP model: late forecasts initialised when the storm had already developed were more accurate than earlier forecasts, with a sharp drop in predictability. A medicane in January 2014 was simulated by different NWP models by Cioni et al. (2016) and their results were found to be strongly affected by the choice of initial conditions.

Our present study details the representation of medicane Ianos, one of the strongest medicanes ever observed (Lagouvardos et al., 2022), in simulations carried out with a state-of-the-art NWP model. A subset of these simulations took part on a model intercomparison project about the predictability of Ianos (Pantillon et al., 2024). In order to explore which processes are more important for the conditioning of the medicane intensification, the simulations are initialised at different times, meaning that some processes conducive to Ianos development are explicitly simulated in simulations with earlier initialisation times, while they are incorporated into the initial state of simulations with later initialisation times. The study also includes simulations with perturbed SST to explore the impacts of enhanced and suppressed sea-surface fluxes. The role of the diabatic processes before and during Ianos's intensification is diagnosed with the use of the Semi-Geotriptic inversion tool (SGT, Cullen 2018), and the role of baroclinic forcing with the use of the Quasi-Geostrophic omega equation (QG, Davies 2015).

Details of our simulations, the SGT inversion tool, the QG omega equation plus other evaluation tools and the observational datasets employed are provided in Sect. 2. A synoptic overview of the Ianos case is given in Sect. 3. The results from the simulations are presented in Sect. 4. The main conclusions of this study are described in Sect. 5.

## 2  Methodology

Regional simulations of medicane Ianos are carried out with the Met Office Unified Model (MetUM) at 2.2 km horizontal grid spacing; see sub-section 2.1 for further details on the MetUM setup. The region chosen for the simulations, the so-called model domain, is a bespoke choice for the Ianos case. The Model's grid-spacing and domain follow a model intercomparison protocol detailed in Pantillon et al. (2024). The region extends from the Tyrrhenian Sea to to the Aegean Sea (Fig. 1) and encompasses the locations of Ianos' cyclogenesis over the Gulf of Sidra, its tropical-like transition over the Ionian Sea and its landfall on the

Greek Ionian Islands. The model is run for five days and initialised every 12 hours from $00Z14$ to $00Z16$ September 2020. Two additional sets of simulations with all the different initialisation times are run with SST increased and decreased by 2 K uniformly over the whole domain (termed +2 K SST and -2 K SST, respectively). The choice of two degrees is motivated by the maximum to minimum range of SST perturbations in the global ensemble system (Tennant and Beare, 2014), it is also an effective way to trigger or suppress near-surface diabatic processes which later create convection and thus maximise or minimise the role of diabatic processes. It does not aim to represent SST changes in the Mediterranean under climate change projections.

The simulations are driven, in terms of both initial data and boundary conditions, by the six-hourly European Centre for Medium range Weather Forecasting (ECMWF) Integrated Forecasting System Analysis (IFS-AN), with the operational resolution and configuration as in September 2020 (cycle 47r1, ECMWF 2020). The IFS-AN is preferred to the fifth generation ECMWF atmospheric reanalysis (ERA5, Hersbach et al. 2020) for its higher resolution: 9 km compared to 25 km. It is also preferred to the internal MetUM analysis as the IFS-AN is more accessible and can be ingested by other regional NWP models, thus allowing our simulations to be reproduced with other models. SSTs are prescribed at the start of the simulation, and are interpolated from the 1/20° Operational Sea Surface Temperature and Sea Ice Analysis (OSTIA, Donlon et al. 2012). The soil moisture analysis comes from the Met Office land-model analysis for consistency; it is described in Gómez et al. (2020).

The dynamical fields of the simulations are evaluated against the IFS analysis described above. The precipitation is compared to the Integrated Multi-satellitE Retrievals for the Global PRecipitation Measurement (IMERG or GPM (Hou et al., 2014; Skofronick-Jackson et al., 2017), abbreviated as GPM hereafter). The GPM spatial and temporal resolution is $0.1° \times 0.1°$ and 30 min, respectively. The satellite product is calibrated with month-to-month gridded gauge data. The representation of precipitation by GPM over the Mediterraean region is considered accurate, but it underestimates precipitation over mountainous regions (Navarro et al., 2019). GPM tends to overestimate extreme precipitation events such as those associated with tropical cyclone Imelda (Sakib et al., 2021) and those above the $95\%$ percentile over the Maritime Continent (Da Silva et al., 2021).

The tropical transition in Ianos is explored with the cyclone phase space diagnostic of Hart (2003), with a couple of differences in its methodolody. First, our domain is small so we use a radius of 400 km instead of 500 km, see section 3.c of Mazza et al. (2017) for a discussion on the choice of different radius. Second, the thickness asymmetry term $B$ is computed using the 925 hPa level instead of 900 hPa, as the latter level is not available in our simulations. Tropical features in Ianos including the cloud-less eye and spiralling branches are explored with the help of the Satellite-like Imagery Simulator (SIMIM), the wrapper to pre-process, run and post-process the unmodified core of the Radiative Transfer for TIROS Operational Vertical Sounder version 12 (RTTOV, Saunders et al. 2018), a fast radiative transfer model for simulating top-of-atmosphere radiances. Cyclone tracks are computed by a simple algorithm that tracks for the minimum pressure over a $4° \times 4°$ box with its centre at the cyclone's previous location. Despite its simplicity, the obtained cyclone tracks and minimum pressures compare well to those from more complex tracking algorithms such as the one used operationally by the Met Office for tropical cyclones (described in Heming (2017)).

Two medium complexity models are employed to understand the importance of diabatic processes, enhanced by surface fluxes induced by warmer SST or suppressed by cooler SST, and upper-level baroclinicity in Ianos's intensification. The role of

diabatic processes is explored with the Semi-Geotriptic inversion tool, which derives the balanced ageostrophic wind solutions from the MetUM fields; the tool is described in further detail in sub-section 2.2. Quantification of the forcing from upper-levels is done via the Quasi-Geostrophic omega equation, detailed in sub-section 2.3.

## 2.1 MetUM

The MetUM is a seamless modelling framework for weather forecasting and climate prediction across scales (Brown et al., 2012). Our simulations use the MetUM regional setup at version 12.0 with a grid spacing of 2.2 km, 90 terrain-following vertical levels with model top at 40 km, a time step of 75 seconds and a regional configuration known as Regional Atmosphere and Land version 2 for the mid-latitudes (RAL2-M), described in Bush et al. (2023). The RAL2M configuration does not include any parametrization for convection, its boundary layer scheme blends a 1-D convection parametrization, which is

suitable for large grid lengths, with a subgrid turbulence scheme suitable for large-eddy simulation (Boutle et al., 2014). These characteristics allow the model to explicitly represent convective events and, in particular small-scale features such as a cloud-less eye.

The main components of RAL2-M are the "Even Newer Dynamics for the General Atmospheric Modelling of the Environment" (ENDGAME) dynamical core (Wood et al., 2014), the single moment micro-physics scheme based on Wilson and

Ballard (1999) with particle size distribution for rain from the rain-rate-dependent distribution of Abel and Boutle (2012) and the snow size distribution parameterization of Field et al. (2007), the "Suite Of Community RAdiative Transfer codes based on Edwards and Slingo" (Socrates) radiation scheme (Manners et al., 2018), the Smith (1990) cloud scheme with empirical adjustments to cloud fraction based on Boutle and Morcrette (2010), the zero lateral flux (ZLF) moisture conservation scheme of Zerroukat and Shipway (2017), and the Joint community land-surface model (JULES, Best et al. 2011). Despite its opera-

tional use in regional models over the UK since December 2019, the RAL2M configuration suffers from heavy precipitation biases, producing too much intense rainfall and too little drizzle (Bush et al., 2023). The use of the two momentum Cloud AeroSol Interacting Microphysics (CASIM Field et al., 2023, ;) reduces some of this bias and will be included in future RAL configurations along with other improvements (Bush et al., 2024).

## 2.2 Semi-Geotriptic inversion tool

The Semi-Geostrophic (SG) balanced flow is derived from the primitive equations after applying the hydrostatic balance and the geostrophic momentum approximation. It is an accurate approximation to the 3-D compressible Euler equations on scales larger than the Rossby radius of deformation (Cullen et al., 2005; Cullen, 2018). The inversion tool is a single timestep integration of the SG system with atmospheric fields produced by a MetUM simulation. Hence, it provides a solution for the MetUM flow consistent with the maintenance of geostrophic and hydrostatic balance. The semi-geotriptic (SGT) model improves the semi-

geostrophic model by the inclusion of Ekman friction in the boundary layer (Beare and Cullen, 2010). In our study, the use of the SGT targets the impact of diabatic processes in the upper troposphere, where there is minimum friction. We thus refer to the ageotriptic wind as the ageostrophic wind, and SemiGeotriptic as Semi-Geostrophic, as these are more familiar concepts in the literature. The SGT inversion tool has been previously applied to quantify the importance of diabatic processes in model

error growth over the North Atlantic (Sánchez et al., 2020). It can also be applied in the tropics using the weak-temperature

gradient, as done in Hardy et al. (2023) to explore the dynamics of synoptic vortices over the South China Sea.

The mathematical formulation of the SGT tool is detailed in Cullen (2018) and Sánchez et al. (2020). Here, it is described by eq. 1, a simpler formulation of the momentum equation combined with geotriptic and hydrostatic balance, derived in Cartesian coordinates, and applying the Boussinesq approximation. The Matrix Q is shown in eq. 2 where $\mathbf{u_g} = (u_g, v_g)$ denotes the geostrophic wind, $\phi$ the geopotential, $\theta$ the potential temperature with $\theta_0$ the environmental value, and $f$ the Coriolis parameter.

Momentum forcing from gravity wave drag, boundary-layer or convection parametrization is represented by the $(M1, M2)$ vector and the diabatic forcing from MetUM parametrizations as $S_\theta$. Note, in this study we neglect the momentum terms ($M1$ and $M2$), in line with Sánchez et al. (2020), and only consider diabatic forcing from the radiation, large-scale precipitation, cloud and boundary layer MetUM parametrizations. As mentioned above, there is no convection scheme in the RAL2M MetUM configuration, so the temperature changes during convective ascents are represented by the other parametrizations. The SGT

system is reduced to an elliptic equation for $\phi$ after replacing the wind vector by the continuity equation under the Boussinesq approximation and applying the $\nabla \cdot \mathbf{Q}^{-1}$ operator to eq. 1. Then the ageostrophic winds are diagnosed by applying the $\mathbf{Q}^{-1}$ operator to eq. 1 and the new value of $\phi$ to obtain its temporal derivative. The full mathematical procedure in spherical and terrain-following coordinates can be found in Cullen (2018).

$$
\mathbf{Q}
\begin{bmatrix}
u - u_g \\
v - v_g \\
w
\end{bmatrix}
+ \frac{\partial}{\partial t}(\nabla \phi) =
\begin{bmatrix}
-f\mathbf{u}_g \cdot \nabla v_g + fM2 \\
f\mathbf{u}_g \cdot \nabla u_g - fM1 \\
-\frac{g}{\theta_0}\mathbf{u}_g \cdot \nabla \theta + \frac{g}{\theta}S_\theta
\end{bmatrix}
\tag{1}
$$

$$
\mathbf{Q} =
\begin{bmatrix}
f^2 + f\frac{\partial v_g}{\partial x} & f\frac{\partial v_g}{\partial y} & f\frac{\partial v_g}{\partial z} \\
-f\frac{\partial u_g}{\partial x} & f^2 - f\frac{\partial u_g}{\partial y} & -f\frac{\partial u_g}{\partial z} \\
-\frac{g}{\theta_0}\frac{\partial \theta}{\partial x} & -\frac{g}{\theta_0}\frac{\partial \theta}{\partial y} & -\frac{g}{\theta_0}\frac{\partial \theta}{\partial z}
\end{bmatrix}.
\tag{2}
$$

The SGT forcing is on the right hand side of eq. 1. The first term represents the geostrophic forcing, and the second term the diabatic forcing coming from subgrid parametrizations. Either of these terms can be set to zero in the SGT inversion tool. For the cases where the SGT is forced by the geostrophic term only, hence $S_\theta = 0$, we term the SGT inversion tool solution $NO - HEAT$. For the cases where the geostrophic forcing is set to zero and only the diabatic term is applied, the solution is

termed $HEAT - ONLY$; this solution represents the balanced flow induced by diabatic heating.

MetUM prognostic fields and tendencies from the physics schemes are output every six hours for the SGT inversion tool. Due to concerns about how the SGT inversion tool will converge with "noisy" vertical motions in the 2.2 km convective-scale model output (as previous use of this tool has used convection-parametrizing global model output), we first filter scales above wave-number 50, equivalent to a wavelength of $0.32°$, following the discrete cosine filtering technique of Denis et al. (2002).

High vertical winds in and around convective cores are thus filtered and are equivalent to the ones in the global model employed by Sánchez et al. (2020) and Hardy et al. (2023), where the SGT inversion tool was successfully applied. Then we follow their

methodology to bi-linearly regrid the fields to a $1.5° \times 1.5°$ latitude-longitude mesh so the SGT inversion tool can accurately solve the semi-geostrophic flow at and above the Rossby radius of deformation (Cullen, 2018) at a lower computational cost.

## 2.3 Quasi-Geostrophic omega equation

The SGT inversion tool currently has no capacity to restrict the forcing terms to certain levels. The Quasi-Geostrophic (QG) omega equation on the other hand allows height-attributable solutions, which have been applied to the classification of extra-tropical cyclones according to the contribution from upper or lower levels in their development (Deveson et al., 2002; Plant et al., 2003; Gray and Dacre, 2006). Moreover the QG omega equation is conceptually simpler as well as more commonly used in the literature, and so it is used here to understand the role of upper-level baroclinicity (without the inclusion of diabatic forcing) in the development of Ianos.

The QG system is based on a first order approximation of the primitive equations with a small Rossby number. It prescribes the ageostrophic vertical velocity required by the prevailing geostrophic flow to maintain geostrophic balance. The QG system is simpler than the SG system: it assumes uniform static stability on each level whereas the SG system allows horizontal gradients of static stability. Additionally the QG system only allows advection of temperature and momentum by the geostrophic wind, whereas the SG system also allows advection by the ageostrophic velocity. For further details on the differences between the SG and QG systems see Sect. 2.3 of Sánchez et al. (2020).

The QG omega equation has been extensively used to examine the large-scale vertical velocity patterns of atmospheric systems. It has several formulations, detailed in Davies (2015). A conventional formulation using height as the vertical coordinate (such that it is an expression for vertical velocity, w, rather than omega), and assuming a constant value of the Coriolis parameter and density and that there is no friction or heating, is shown in eq. 3.

$$N^2 \nabla_H^2 w + f^2 \frac{\partial^2 w}{\partial^2 z} = f \frac{\partial}{\partial z} \left[ v_g \cdot \nabla \zeta_g \right] - \frac{g}{\theta_0} \nabla_H^2 \left[ v_g \cdot \nabla \theta \right] \tag{3}$$

Where $N$ is the buoyancy frequency, $\nabla_H$ the horizontal gradient and $\zeta_g$ is the geostrophic relative vorticity. The first term on the right-hand-side is commonly referred as the geostrophic Vorticity Advection term (VA), and the second as the Thermal Advection term (TA). Forcing from VA is assumed to dominate the QG forcing at upper levels yielding the upper-level forced ascent (Deveson et al., 2002). The QG inversion tool cannot achieve convergence with MetUM fields on their 2.2 km native grid. Hence, the MetUM horizontal winds, $\theta$ and $\phi$ fields are spectrally truncated to wave-number 16, equivalent to $1°$, using the Discrete Cosine Function (DCT) of Denis et al. (2002). In order to minimise the Gibbs phenomenon, the DCT includes the smoothing spectral filter of Sardeshmukh and Hoskins (1984).

## 3 Case Description

Medicane Ianos travelled from the Gulf of Sidra to the Greek Ionian islands between 15 and 18 of September 2020. Ianos started its cyclogenesis on 15 September over anomalously warm waters at 28°C, 1.5 K to 2.0 K warmer than the 1985–2005 climatology (Lagouvardos et al., 2022; Zimbo et al., 2022). The environment was dominated by a cut-off low above the Gulf of

Sidra, isolated from the Atlantic flow on the 11[th], and a series of intense but disorganised co-located convection events on the
13[th] and 14[th]. Ianos intensified and attained a full tropospheric warm core at 03Z on 17[th] (denoted as $03Z\,17$, and similarly for
other times, hereafter), as shown in CPS diagrams in Zimbo et al. (2022). It initially travelled northwards on the 16[th] across the
Ionian Sea towards Southern Italy, and eastwards afterwards till landfall over the Greek Ionian Islands at $06Z\,18$ with maximum
wind gust of 50 m/s recorded at Palliki station in North Kefalonia and daily accumulated rainfall of 644.7 mm in Antipata, at
the northernmost part of Kefalonia. Diakakis et al. (2023) records Ianos's impacts on geomorphology and infrastructure on the
Ionian islands. On the following days after landfall, Ianos headed south and started to weaken; it lost its tropical characteristics
around $00Z\,19$. Ianos's track in the IFS analysis is shown in Fig. 1. Table 1 briefly describes Ianos's evolution.

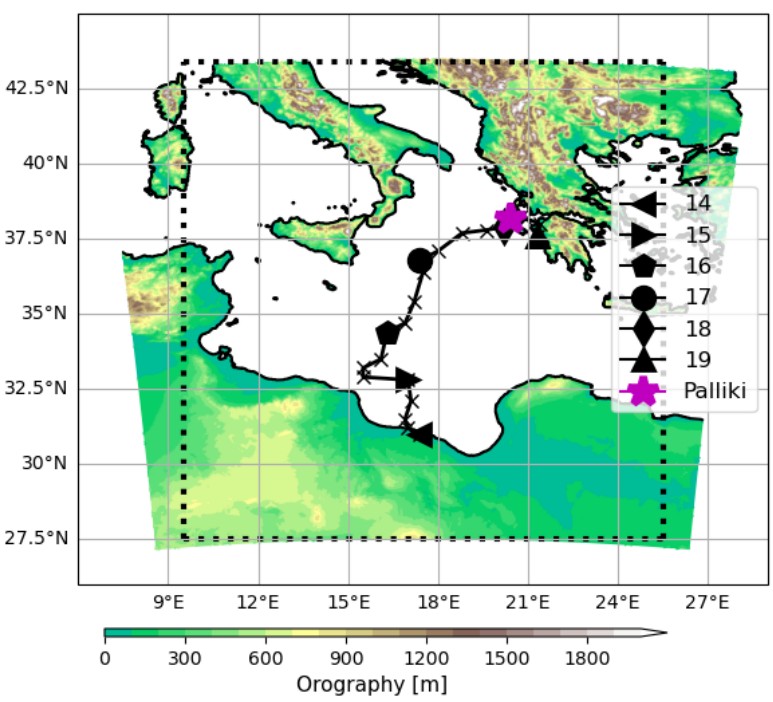

**Figure 1.** Domain for the 2.2km MetUM simulations and its orography (coloured). The black dotted box represents the area where Ianos is
evaluated in the MetUM simulations. Ianos's track from the IFS analysis is represented by the black line, with crosses every six hours and
different symbols for different days in September 2020 at $00Z$; only points with surface pressure below 1012 hPa are shown. The magenta
asterisk denotes the location of Palliki Station in Kefalonia.

| Day | Description |
|---|---|
| 11 to 15 | Strong convective precipitation in the Gulf of Sidra below a cut-off low. |
| 15 | Ianos's cyclogenesis over the Gulf of Sidra with 28°C SST, 2 K above climatology. |
| 16 | Ianos moves north while it intensifies. PV streamer wraps around the cyclone on the 335K isentrope at 12Z. |
| 17 | Ianos develops a warm core and an eye with intense convection and wind gusts around the eye. Ianos moves eastwards towards the Greek Ionian Islands. |
| 18 | Ianos hits the Greek Ionian Islands in western Greece: daily accumulated precipitation peaks above 600 mm and wind gusts above 50 m/s. |
| 19 | Ianos moves south towards Levant and decays, losing its tropical-like features |

**Table 1.** Brief timeline of the development of Ianos and its preceding events in September 2020.

## 4  Results

Ianos's intensity and tracks in the analyses and simulations are described in Sect. 4.1, the convection preceding Ianos and during its intensification are in Sect. 4.2, and the development of Ianos's tropical features are in Sect. 4.3. A description of the upper-level flow in the IFS analysis and in the different simulations are in Sects. 4.4 and 4.5, respectively. The impact of the diabatic heating in such flow from the SGT inversion tool is in Sect. 4.6 and the baroclinic forcing via the QG inversion tool in Sect.4.7.

### 4.1  Ianos's track and intensity

The minimum Mean Sea Level Pressure (MSLP) along Ianos's track from analyses and MetUM simulations is shown in Fig. 2, with the minimum pressure observed at Palliki station in Kefalonia (Greece) at landfall reported in Lagouvardos et al. (2022). The cyclone starts to intensify in the analysis at $00Z15$ and at a faster rate from $00Z16$. It peaks on the 17[th] when it develops a deep warm core (see Sect. 3), and then it starts to decay sharply after landfall at $06Z18$. The IFS and MetUM analyses are 5 hPa and 10 hPa weaker, respectively, than the observed value at Palliki station at $03Z18$, despite the location offset of the station to the cyclone's minimum pressure location. Tropical cyclone warning centres around the globe (e.g., the Japan Meteorological Agency and the National Hurricane Center) produce estimates of the intensities of cyclones which are assimilated by models such as the IFS (Heming, 2016). However, there is no official warning centre for Mediterranean cyclones and thus the weaker intensities in the analyses could be the result from the lack of central pressure estimates.

All the five simulations with +2 K SST develop a strong Ianos (copper lines of different shadings and symbols in Fig. 2), reaching a minimum MSLP of 945–960 hPa. For the subset of simulations with control SST, only those initialised at and after $00Z15$ capture the intensification of Ianos (green lines with circles, diamonds or crosses in Fig. 2), with a range of maximum intensities of 970–985 hPa. There is only one simulation with -2 K SST capturing Ianos's intensification albeit weakly, the one initialised at the latest time: $00Z16$ (pale blue line with crosses in Fig. 2). The maximum intensity of Ianos is higher in the +2 K SST simulations initialised at the three earliest times than in the latest two, probably the result of having the extra

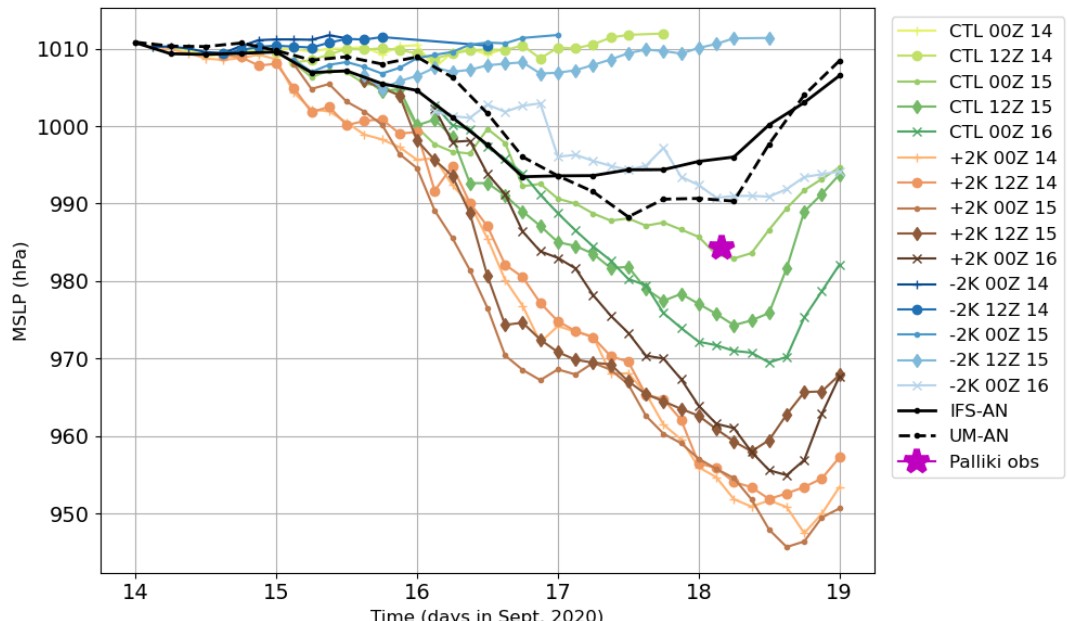

**Figure 2.** Ianos's MSLP evolution from 14–19 September for the IFS analysis (black dashed), the MetUM analysis (black dotted), and all MetUM simulations (green for control, copper for +2 K SST and blue for -2 K SST). Simulations initialised at an earlier time are represented by darker colours and different symbols. The magenta star denotes the observed minimum MSLP at the Palliki Station in Kefalonia. Only points below 1012 hPa are shown. The first point in the simulations corresponds to a three-hour leadtime forecast (denoted T+3).

source of surface heat and moisture fluxes for a longer duration. Conversely, the control simulations able to simulate Ianos produce stronger intensities for later initialisation times (shorter lead times). The lack of cooling feedback in these prescribed

SST simulations may produce an over-intensification of Ianos, as is common with tropical cyclones in simulations uncoupled to ocean models (e.g Short and Petch 2018; Castillo et al. 2022).

The track differences amongst simulations are less structured than those for the intensities. Their differences to the IFS analysis in terms of latitude and longitude are shown in Fig. 3. In the simulations where Ianos does not intensify and remains a rather weak mid-latitude low (above 1008 hPa), it does not travel as far north as it does in the simulations where it rapidly

intensifies; e.g., the control simulation initialised at $00Z\,14$ (olive-green pluses) and $12Z\,14$ (olive-green circles), or the simulation with -2 K SSTs initialised at $12Z\,15$ (pale-blue diamonds). Most of the simulations show a $1°$ western bias emerging on the 17[th], indicating that the cyclone is slower in the simulation than in the analysis, a positional bias leading to a later landfall. This bias gives Ianos more time to be fed by sea-surface fluxes, which may explain why Ianos's minimum surface pressure is reached about one day later in the simulations than in the analyses (Fig. 2).

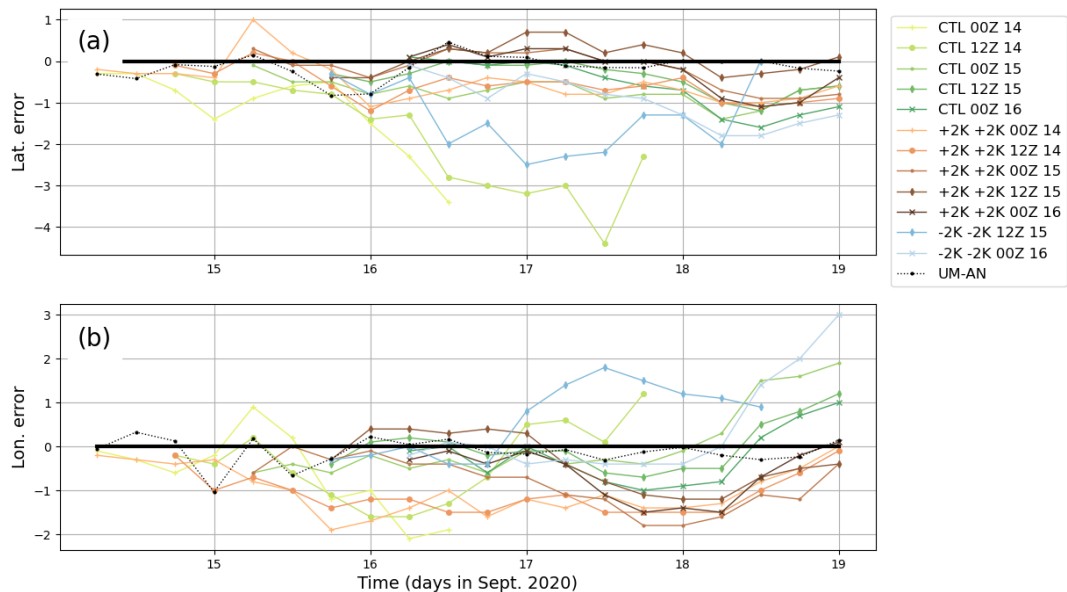

**Figure 3.** Ianos's track errors compared to the IFS analysis for the different simulations in (a) latitude (b) longitude. Same colour legend as in Fig. 2.

## 4.2 Evolution of Precipitation

Heavy precipitation occurred over the Gulf of Sidra and the southwest sector of the Ionian Sea before Ianos's cyclogenesis. It was of convective nature and quite disorganised despite being fairly intense (Zimbo et al., 2022). One possible mechanism for the thunderstorm development is so-called "dry-eyes", where stratospheric intrusions into the lower troposphere are shown as isolated dark spots on the water vapour channel of the satellite imagery, triggering potential instability and thus convection (Roberts, 2000). Dry eyes were observed over the region on the 13[th] in the water vapour channel ($6.25\mu m$) by the Spinning Enhanced Visible and InfraRed Imager (SEVIRI) instrument on board of the Meteosat Second Generation Satellite (MSG, not shown but available at the CEDA archive (Met Office and EUMETSAT, 2006)).

The satellite-derived product GPM shows intense precipitation over the Gulf of Sidra between $12Z\,14$ and $12Z\,15$ (Fig. 4.a). The precipitation location over this period is spatially collocated with Ianos's track downstream between $00Z\,15$ and $06Z\,16$ (right-ward arrow and pentagon symbols in Fig. 4.a). From all three simulations initialised at $00Z\,14$, only the one with +2 K SST captures the location of the preceding rainfall downstream of the cyclone (Fig. 4.b). The simulation with control SST initialised at $00Z\,14$ produces much weaker accumulated precipitation over the same time and location as GPM (Fig. 4.c), but slightly stronger precipitation over the Tunisian coast. Ianos does not travel over the area with preceding precipitation until $00Z\,16$, a day later than when it occurs in GPM or the equivalent simulation with +2 K SST. The -2 K SST simulation has

nearly absent precipitation over the Gulf of Sidra (Fig. 4.d), probably the result of weaker surface fluxes reducing the thermal vertical gradient and suppressing convective activity over the sea.

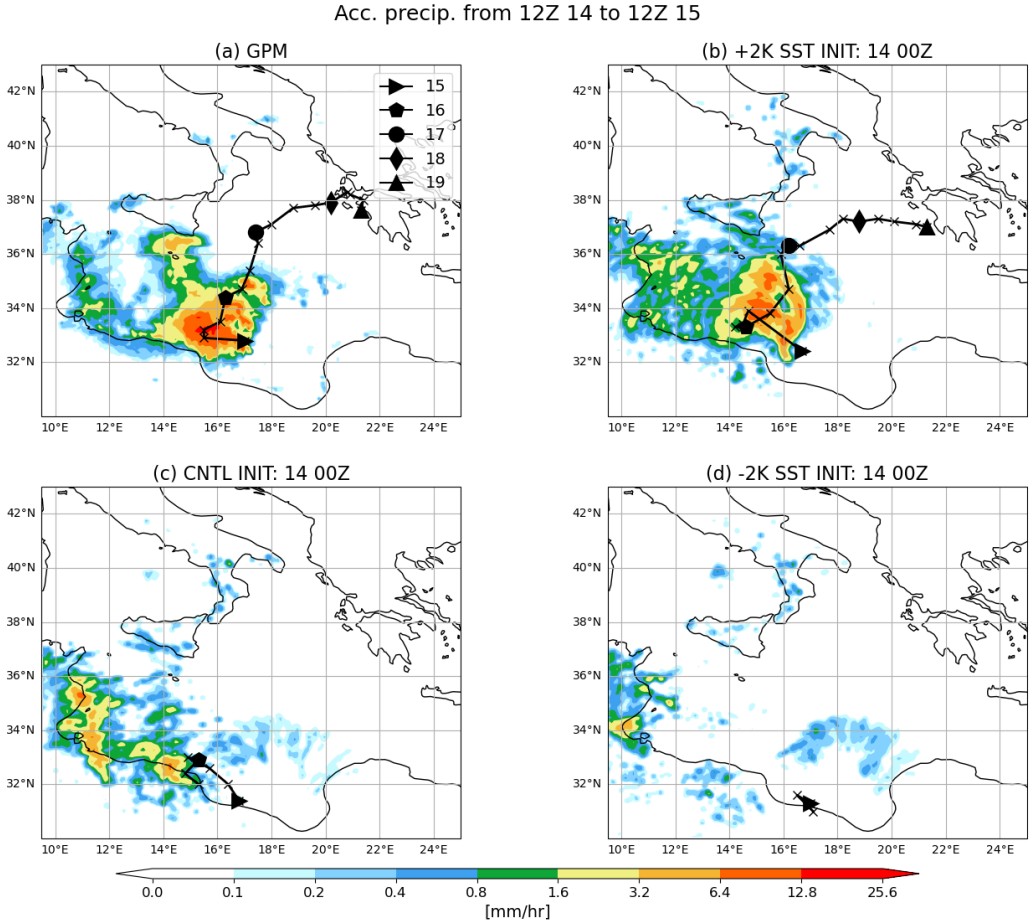

**Figure 4.** Daily accumulated precipitation from $12Z\,14$ to $12Z\,15$ for GPM and simulations initialised at $00Z\,14$; (a) GPM, (b) control (c) +2 K SST (d) -2 K SST. Black line denotes the cyclone tracks of the IFS analysis in (a) and simulations in (b–d) with crosses every six hours and different symbols for different days in September 2020 at $00Z$ from $00Z\,15$; only values below 1012 hPa are shown.

The evolution of six-hourly accumulated precipitation along the cyclone's track is shown in Fig. 5 as the time-series of the domain averaged precipitation over a $3\times3°$ box with the cyclone at the centre. The GPM satellite product shows two maxima in precipitation around the cyclone. The first at $00Z\,15$ is the preceding precipitation event over the Gulf of Sidra shown in Fig. 4, when Ianos is just a weak small perturbation in the surface pressure field; this peak is refereed to as preceding precipitation or peak "A" hereafter. The second maximum is at $12Z\,16$ when Ianos is undergoing a rapid intensification; it is refereed to as

Ianos's intensification precipitation peak or peak "B" hereafter. Other medicanes also show the equivalent to Ianos's peak "B", producing intense precipitation, lighting and deep convection 12–24 hours ahead of achieving medicane status (Miglietta et al., 2013; Fita and Flaounas, 2018).

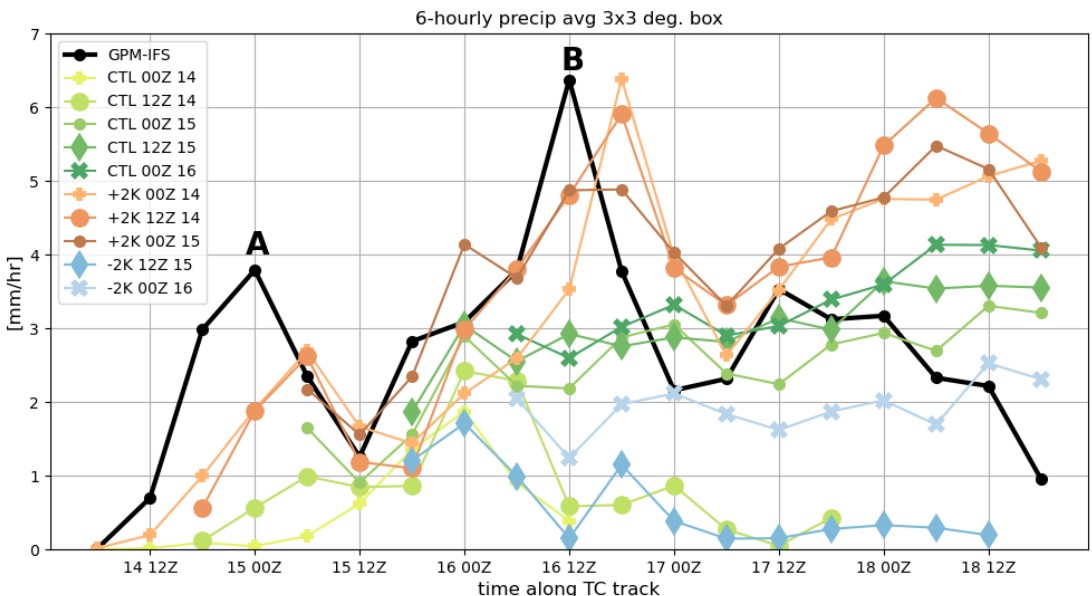

**Figure 5.** Timeseries of six-hourly precipitation along Ianos's track, averaged over a $3 \times 3°$ box with the cyclone at the centre and using the same colours and symbols as Fig. 2. A few simulations have been removed for clarity.

The simulations with +2 K SST are the only ones capturing both precipitation peaks, although they also develop a spurious third peak at $12Z\,18$ (Fig. 5). This third peak comes from a later landfall and thus a longer exposure to the warmer SST (Figs 3 and 4.a,b). The simulations with +2 K SST initialised at $12Z\,15$ and $00Z\,16$ follow a similar pattern to the ones initialised at earlier times and are not shown. The precipitation timeseries in the simulations with control SST initialised on the 14th miss peak "A" (olive-green lines with circles and pluses in Fig. 5), but in these simulations the precipitation increases at about the

same rate on the 15th as the other simulations with control SST which are able to develop Ianos. However, in those simulations missing Ianos's intensification, the precipitation rate drops after reaching its maximum early in the 16th, whereas in those simulations initialised afterwards, and capturing Ianos intensification, there is ongoing precipitation after peak "B" (Fig. 5). The simulations initialised at and after $12Z\,15$ follow a similar pattern to the one initialised at $00Z\,15$ and are not shown. Most of the simulations with -2 K SST produce negligible precipitation with the exception of the ones initialised at the latest two

290  times (not shown). Only the latest one (initialised at $00Z\,16$) produces medicane Ianos and substantial precipitation along its track, whereas the one initialised 12 hours earlier (at $12Z\,15$) produces a drop in the intensity and precipitation within a few

hours (blue line with diamonds in Figs. 2 and 5 respectively). The moisture availability in the lower levels is substantially reduced compared to the control simulations due to the small values ($< 150 W/m^2$) of latent heat fluxes (not shown).

Only the simulations with warmer SST, hence stronger surface fluxes leading to enhanced convective activity, are able to simulate the precipitation event before Ianos's intensification, and its intensification. Once the dynamical feedbacks from the preceding precipitation are in the analysis, from $00Z\,15$ onwards, the simulations with control SST are able to capture Ianos's intensification as well. However, simulations with -2 K SST initialised on the 15[th] kill the cyclone development. Hence the dynamical feedbacks from the preceding precipitation triggering Ianos's intensification can be cancelled by weaker surface fluxes. These results highlight the importance of the preceding precipitation event in conditioning the environment to become more favourable for Ianos's intensification.

### 4.3 Ianos's tropical transition

Wind shear at the time of cyclogenesis is between 10 to 15 m/s in the IFS analysis and all simulations and increases uniformly thereafter (not shown) with negligible differences between simulations developing and not developing medicane Ianos. The environment where Ianos's cyclogenesis occurs on the 15[th] after the preceding convection is characterised by a weak thermal wind shear and horizontal symmetry, as shown by the timeseries of the cyclone phase space variables in Fig. 6.

When Ianos begins to intensify on the 16[th] it moves out of the preceding convection area and into an environment with less symmetric low-to-mid level thickness, hence $B$ increases (Fig. 6.a). About one day later $B$ begins to decrease and Ianos develops a symmetric warm core after $06Z\,17$, coinciding with the date reported in Lagouvardos et al. (2022). The thermal wind increases in the upper and lower troposphere while Ianos intensifies, which implies that the warm core strengthens, even after it develops an axisymmetric core, in contrast to the IFS analysis (Fig. 6.b,c). The thermal wind at the upper and lower levels is stronger in the simulations with +2 K SST, but it is not consistently stronger in those simulations with a more intense Ianos. The simulations not developing medicane Ianos shown in Fig. 6 are the ones with control SST initialised at $12Z\,14$ and the one with -2 K SST initialised at $12Z\,15$. Neither of these two simulations develop an upper-level warm core nor an axisymmetric core.

The tropical-like cloud-less eye and spiralling cloud bands can be observed in the simulated satellite Brightness Temperature ($Tb$) at 10.8 $\mu m$ (Fig. 7), from the satellite-like imagery simulator SIMIM in the simulations initialised at the latest initialisation date, $00Z\,16$. For reference, Zimbo et al. (2022) observed $Tb$ 216 K at $03Z\,18$ (equivalent to the lighter white colours in Fig. 7). The simulation with +2 K SST generates the largest and strongest Ianos (Fig. 7.a), with a diameter of approximately 500 km, with the lowest $Tb$ and thus highest cloud tops. In the simulation with -2 K SST Ianos still develops an axisymmetric warm core (Fig. 6). The imagery simulator shows the eye and spiralling branches (Fig. 7.c), but they are much smaller in size and the higher $Tb$ indicates weaker convective activity.

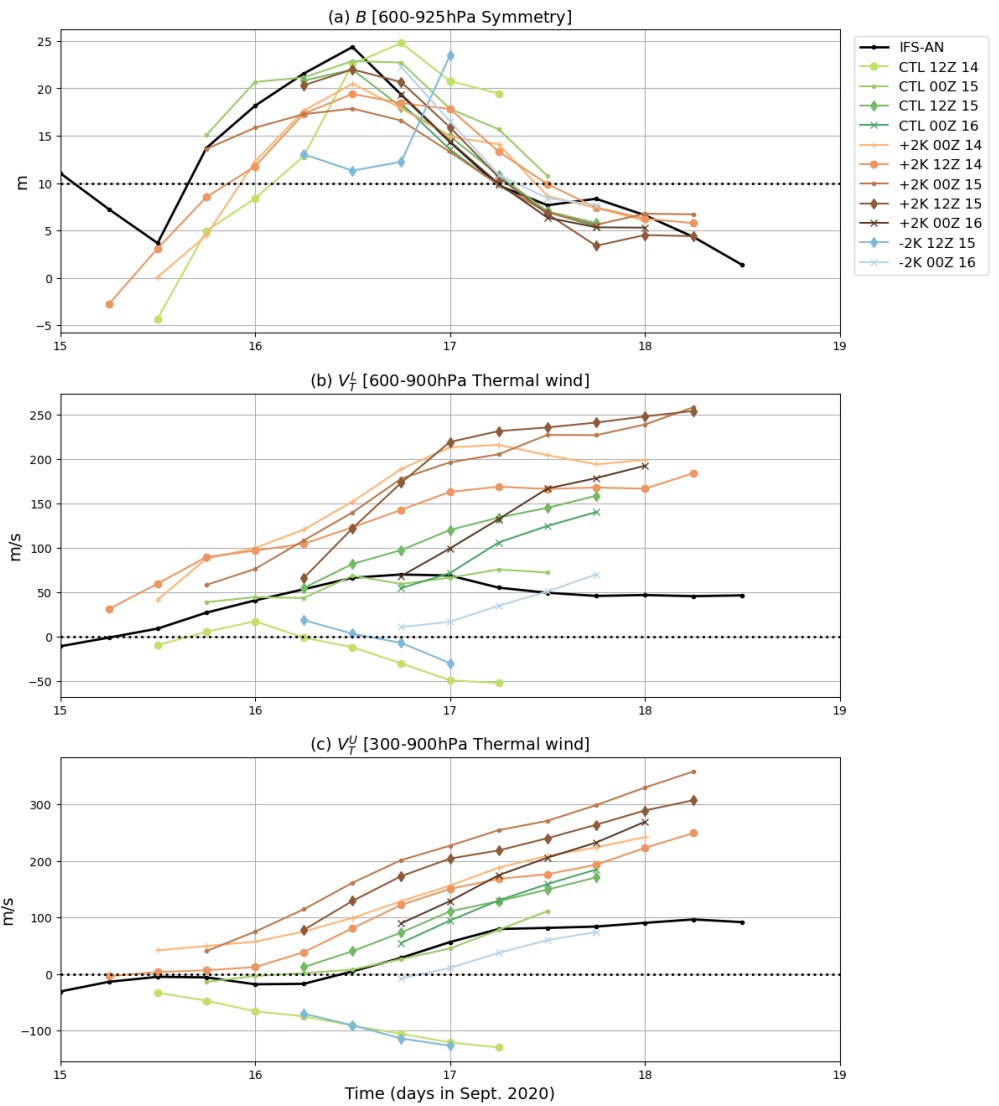

**Figure 6.** Time-series of (a) $B$ term: thickness symmetry between 600–925 hPa, (b) $V_T^L$: low-level thermal wind between 600–900 hPa and (c) $V_T^U$: upper-level upper thermal wind between 300–600 hPa. The dashed black line indicates the thresholds for tropical transition. Values of $B > 10$ shows an anti-symmetric (frontal) core and $B < 10$ a symmetric core. Values for $V_T^L$ and $V_T^U$ above zero denote a warm core and below a cold core. Only points with a circle of 400km radius around Ianos inside the domain are shown, hence points too south at the beginning of the simulations are not shown.

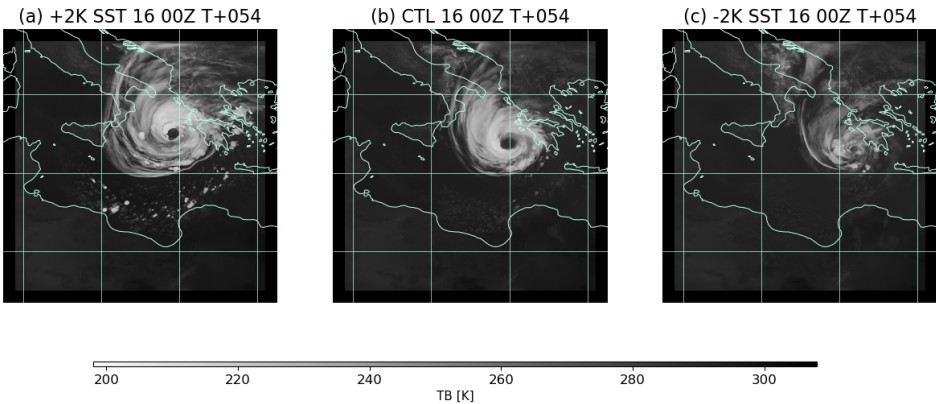

**Figure 7.** Brightness Temperature from the SIMIM satellite simulator at $10.8~\mu$m for the validation time $06Z\,18$: (a) control, (b) +2 K SST, and (c) -2 K SST simulations, all initialised at $00Z\,16$.

### 4.4 Evolution of the upper-level structure and associated PV-tower in the IFS analyses

An analysis of the upper-level fields, in particular of the PV and the horizontal winds, is performed in this sub-section to explore the baroclinic environment during Ianos's life time and the preceding precipitation event. Our definition of upper levels

is the vertical average between 250 hPa and 200 hPa, the range where the maximum wind speed of the upper-level jet is found.

The IFS analysis at $12Z\,14$ has a trough extending over the Mediterranean region with stratospheric PV values ($PV > 2\,PVU$), shown in Fig. 8.a. Inside the trough, there is a small region of tropospheric PV values ($PV < 2\,PVU$) at 15°E, 36°N. This low-PV region could be the result of either vertical advection of low-PV via cross-isentropic transport as well as adiabatic transport or from the mixing of tropospheric convective up-drafts generating low-PV values on the left (and high PV

values on the right) of the wind shear vector as in Hitchman and Rowe (2017), Harvey et al. (2020) and Oertel et al. (2020). Although distinct PV dipoles are not evident in Fig. 8.a, small-scale PV dipoles appear in the upper levels at the earlier stages of convection in the IFS analysis and MetUM simulations (not shown). The small surface pressure perturbation that will later develop into medicane Ianos (indicated by the single enclosed of 1010 hPa MSLP contour) sits underneath the jet at 17°E, 32°N. One day later at $12Z\,15$, immediately after the heavy precipitation event over the Gulf of Sidra (peak "A" in Fig. 4.a),

the area of low-valued PV has grown in size and has turned into a "low-PV bubble" inside the trough (Fig. 8.b). At the surface, Ianos is located at 16°E,33°N on the southern side of the low-PV bubble. With a surface pressure of 1007 hPa, it has just begun to intensify with an emerging cyclonic circulation at low levels (not shown). At $12Z\,16$, the time of maximum cyclone intensity growth and precipitation around the cyclone, the low-PV bubble has expanded northwards and is associated with a narrow stratospheric PV streamer on its south side, where Ianos is situated (Fig. 8.c). There is a strong southerly flow on the

western side of the low-PV bubble, with an anticyclonic curvature near the cyclone. The irregular shape of the low-PV bubble,

with a protuberance of stratospheric PV over the Aegean Sea, suggests that there could be two bubbles merged: a northern one emerging from the preceding precipitation and the southern one from Ianos's outflow. Twenty four hours later, when the cyclone has developed an axisymmetric warm core at $12Z\,17$, the southern branch of the PV streamer cyclonically wraps up around the location of Ianos's MSLP centre (Fig. 8.d).

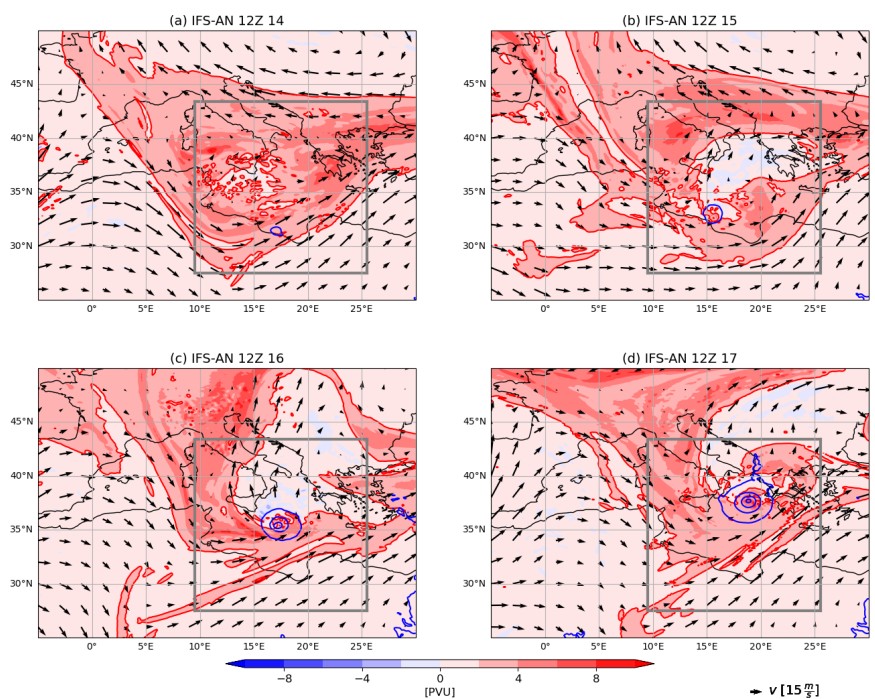

**Figure 8.** IFS analysis PV (colours) and winds (vectors) averaged between 250 hPa and 200 hPa and MSLP (dark-blue contours from 1010 hPa downwards every 5 hPa) at different times: (a) $12Z\,14$, (b) $12Z\,15$, (c) $12Z\,16$, and (d) $12Z\,17$. Red line indicates the dynamical tropopause contour at $PV = 2\,PVU$, where $1\,PVU = 10^{-6}\,Kkg^{-1}m^2s^{-1}$, Grey box denotes the region where the MetUM simulations are evaluated.

For a clear reference to the different stages of Ianos's vertical development, the vertical-latitudinal cross section through the cyclone is shown in Fig. 9, showing the same times as Fig. 8. At $12Z\,14$ the trough stretches between $30°N$ and $45°N$ with southwesterly flow on its southern side and easterly flow on its northern side (Fig. 9.a). The weak pressure perturbation that will later develop into medicane Ianos is located at $32°N$ and denoted by a dark-blue 'x' at the 950 hPa level. At $12Z\,15$ the tropopause is lifted to the 350 K isentropic level between 33 and $38°N$ approximately (Fig. 9.b). This feature is the low-PV

bubble shown in Fig. 8.b. At the same time, the surface signature of Ianos sits underneath the south side of the low-PV bubble at $33°N$. There is diabatic activity right above this location, indicated by the vertical filaments of $PV > 2\,PVU$, which are

possibly associated with the release of latent heat from the preceding convection event. The vertical-latitudinal cross section at $12Z\,16$ shows positive PV production in the vertical at the location of the cyclone at 35°N and negative PV aloft (Fig. 9.c). Steady convective ascent along the column generates diabatic heating, e.g. via latent heat release, and thus generates positive

PV anomalies that extend along the column as shown in Fig 4 of Wernli and Davies (1997), the so-called PV-tower. This PV-tower is a feature of mature cyclones in which low-level positive PV anomalies driven by diabatic processes grow and align with the PV anomaly associated with upper-level troughs (Čampa and Wernli, 2012). There are another two noteworthy results at this time: (i) the PV streamer south of the cyclone intrudes below 300 hPa and (ii) the southerlies above the PV column, which could be driven by the cyclone's outflow. At $12Z\,17$ Ianos has fully developed an axi-symmetric warm core (Fig. 6):

the positive PV anomaly in the low and mid-levels connects with the upper-level stratospheric intrusion (Fig. 9.d). PV towers are also a feature of tropical transitions of cyclones. For example, the composite of cyclones undergoing tropical transition of Galarneau et al. (2015) shows a better-defined PV tower extending to higher levels in the cases developing tropical transition than the non-developing cases. Such a troposphere-deep, positive PV anomaly is also a key feature of tropical cyclones (Thorpe, 1985).

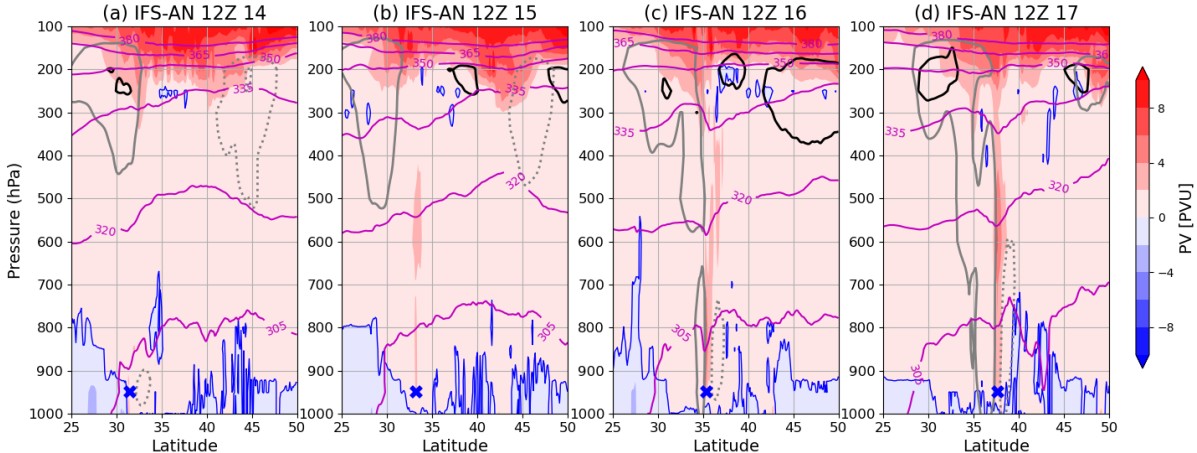

**Figure 9.** Vertical cross sections across latitude for the IFS analysis showing PV (colours), $\theta$ (magenta contours), meridional winds (black contours with a solid line for southerlies and dashed line for northerlies) and zonal wind (grey contours with a solid line for westerlies and dotted line for easterlies) for different times: (a) $12Z\,14$, (b) $12Z\,15$, (c) $12Z\,16$, and(d) $12Z\,17$ (same times as Fig. 8). Wind contours are at 15 m/s. The blue contour shows the $PV = 0\,PVU$, thus enclosing the negative PV areas. The dark-blue cross at 950 hPa denotes the latitude of the cyclone at the time. The location of the cross sections are taken at the longitude of the cyclone and they are longitudinally averaged between $1°$ west and east of the cyclone longitude.

A PV streamer cyclonically wrapping up above the cyclone is a common feature of medicanes right before they develop an axisymmetric warm core (Homar et al., 2003; Miglietta et al., 2011), and intense Mediterranean cyclones before their

maximum intensity (Flaounas et al., 2015, 2021). The PV streamer south of the PV tower intrudes into the troposphere to levels below 300 hPa, as shown in Fig. 9.c,d and wrapping Ianos on the 335K isentropic level as shown in Fig. 6 of Lagouvardos et al. (2022). Yanase et al. (2023) describes the cyclone wrap up for the tropical transition of tropical cyclone Kirogi as an intermediate phase between a baroclinic phase, with a southward upper-level cold trough and a northward low-level warm moist air, and a convective phase with a developed deep warm-core convective vortex. The novelty in the results presented here is that the location and width of the PV streamer is demonstrated to be heavily influenced by the development of a "low-PV bubble", which emerges inside the trough after the preceding convective events.

## 4.5 Upper-level structure in the MetUM simulations

The location and timing of the low-PV bubble and the PV streamer on its south side condition the baroclinic environment where medicane Ianos intensifies. Our family of MetUM simulations are well suited to explore this environment. The simulations with earlier initialisation times allow the dynamical features in the upper levels to develop internally in the model, and those with later initialisation times have these dynamical features included in the analysis. Additionally, our simulations with modified SST overemphasise the effect of enhanced latent heating by stronger surface fluxes (+2 K SST) or reduce diabatic activity with weaker surface fluxes (-2 K SST). Here we analyse the simulations at two different validation times: the end of the preceding precipitation at $12Z15$ and one day later, at $12Z16$, the time of the maximum intensification rate of medicane Ianos.

There are notable differences in the upper-level flow at $12Z15$ across the different simulations initialised at $00Z14$ (Fig. 10, see also Fig. 8.b for a comparison with the IFS analysis). The simulation with 2 K warmer SST shows a wider low-PV bubble and a stronger anticyclonic flow on its western side than the IFS analysis (Fig. 10.a). Ianos is already present at this time in the simulation as a moderately deep cyclone of 1000 hPa (as opposed to a shallow minimum at 1007 hPa in the IFS analysis), and it is located beneath the low-PV bubble at 33°E, 14°N. The simulation with control SST shows an emerging low-PV bubble which has not fully coalesced into a single entity of near-zero PV as in the analysis (Fig. 10.b). It has a more zonal flow around the low-PV pockets. The simulation with -2 K SST has a marginal low-PV bubble with no impact on the cyclonic flow (Fig. 10.c). There is a clear influence of surface fluxes enhancing diabatic activity and fostering the development of the low-PV bubble on the simulation with +2 K SST, and suppressing its development with -2 K SST. The simulation with +2K SST is the closest to the analysis of the three in Fig. 10, despite the strength of Ianos and extent of the low-PV bubble.

From the three simulations initialised at $00Z14$, only the one with +2 K SST develops medicane Ianos. It is also the only one of the three showing a wide low-PV bubble with an intensifying Ianos on its southern side a day later at $12Z16$ (Fig. 11.a). The anticyclonic curvature of the southerly flow on the west-side of the low-PV bubble is clearer than in the IFS analysis (Fig. 8.c), but less marked than one day earlier in the same simulation (Fig. 10.a). The simulation with control SST shows no Ianos at $12Z16$. This simulation has a weak low-PV bubble that is further downstream and has no anticyclonic southerlies (Fig. 11.b). There is no low-PV bubble, Ianos or anticyclonic southerlies in the simulation with -2 K SST (Fig. 11.c).

All the simulations with +2 K SST show a broadly similar flow at $12Z16$, with a low-PV bubble, a PV streamer on its southern side, where Ianos is situated while intensifying at this maximum rate, and anticyclonic southerlies on the north-western side of the cyclone (Fig. 11.a,d,g,j,m). Simulations with the three earliest initialisation times show stronger features,

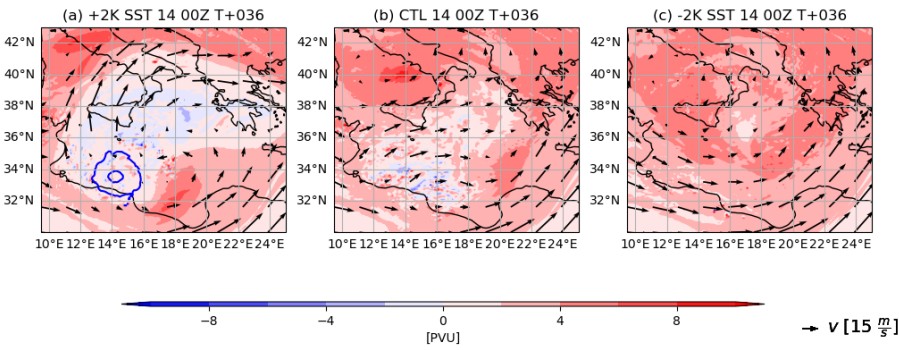

**Figure 10.** Upper-level fields as shown in Fig. 8 for a validation time of $12Z\,15$ and for MetUM simulations initialised at $00Z\,14$ for the (a) +2 K SST, (b) control SST and (c) -2 K SST simulations.

e.g., deeper cyclones (copper lines with pluses, small-dots and large-dots in Fig. 2) and a wider low-PV bubble with higher curvature of the anticyclonic southerly flow (e.g., going west of Sicily in Fig. 11.a,d,g). The earliest two simulations with -2 K SST do not develop a low-PV bubble (Fig. 11.c,f). The next two develop a low-PV bubble, but it is either small (Fig. 11.i) or further downstream of a weak Ianos of 1008 hPa (Fig. 11.l). Only the simulation initialised at the latest time captures the
cyclone intensification, albeit weaker (Fig. 11.o). This simulation also shows the anticyclonic southerlies on the northwest sector of the cyclone at $15°$E, $37°$N.

The experiments with control SST at the five different initialisation times have two clear-cut groups. The first group contains the simulations initialised at the latest three times that are able to develop medicane Ianos. These simulations show an anticyclonic flow emerging from the northwest side of the cyclone (Fig. 11.h,k,n), albeit with a smaller curvature than the
equivalent +2 K SST simulations (e.g., Fig. 11.g and 11.h). The second group of simulations with control SST is composed of the simulations initialised at the two earliest times, in which Ianos either does not exist (Fig. 11.b) or does not intensify and remains as a weak surface low of 1010 hPa upstream of the low-PV bubble (Fig. 11.e). Those simulations are unable to develop Ianos, but they do produce low-PV bubbles. In fact the low-PV bubble shape is very similar between the last simulation unable to develop Ianos, initialised at $12Z\,14$ (Fig. 11.e) and the first simulation able to develop it, initialised at $00Z\,15$ (Fig. 11.h).
The only perceptible difference is the anticyclonic flow and negative PV on the northwest sector of the cyclone at $16°$E, $36°$N in the simulation initialised at $00Z\,15$. Despite these small differences in the upper-level flow, Ianos in the former simulation is a weak surface low of 1010 hPa lagging upstream of the low-PV bubble, whereas Ianos in the later simulation is anchored on the southern side of the bubble and has already deepened below 1000 hPa.

The simulations with +2K SST initialised at the earliest two initialisation times are closer to the IFS analysis than the
control equivalents. However, for the simulations initialised at $00Z\,15$ the simulation with control SST is slightly better than

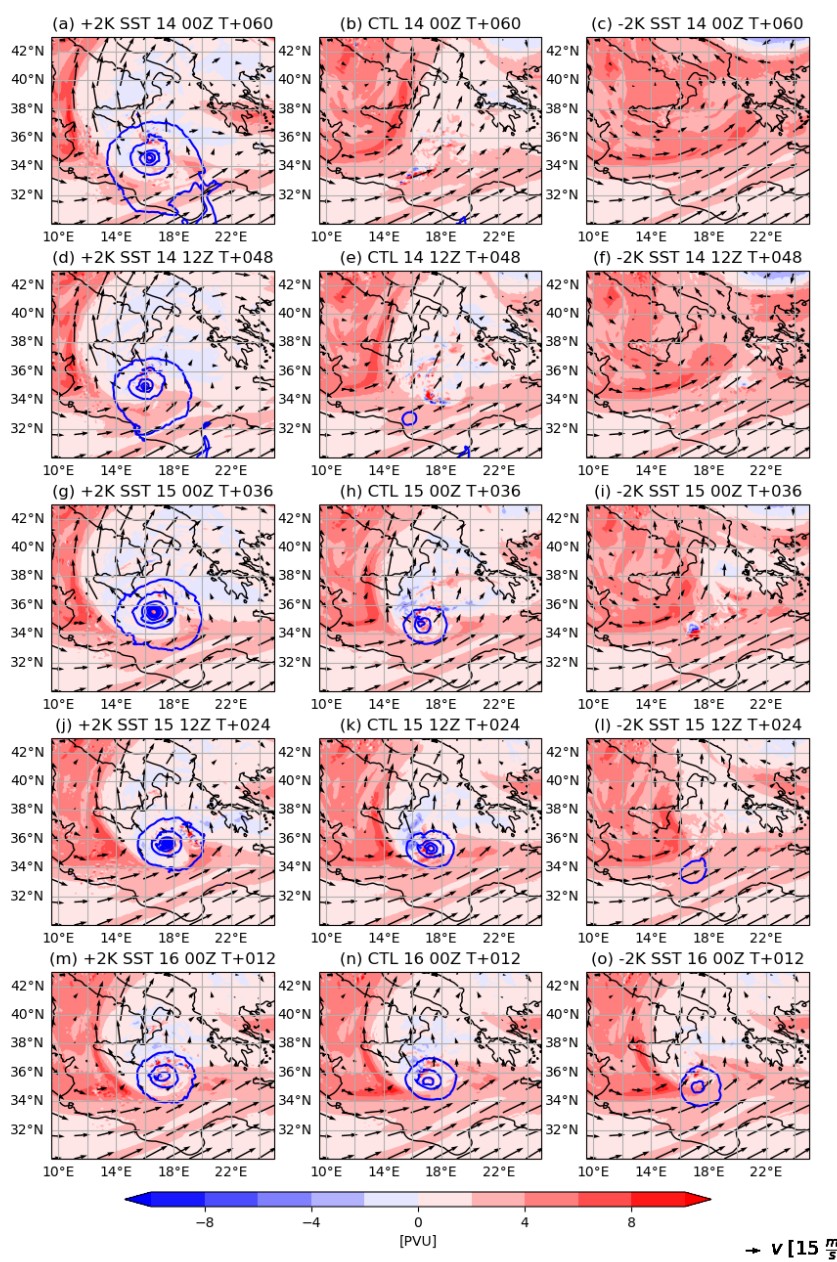

**Figure 11.** Upper-level fields as shown in Fig. 8 for a validation time of $12Z\,16$ and MetUM simulations initialised at (a,b,c) $00Z\,14$, (d,e,f) $12Z\,14$, (g,h,i) $002Z\,15$, (j,k,l) $12Z\,15$ and (m,n,o) $00Z\,16$; with (a,d,g,j,m) +2 K SST, (b,e,h,k,n) control and (c,f,i,l,o) -2 K SST.

the one with +2K SST, as the low-PV bubble in the latter is too wide. For latter two initialisation times, the low-PV bubbles are comparable to the one in the IFS-analysis in the simulations with control and +2K SST. The analysis of the upper-level dynamics reveals three different features in the simulations able to develop medicane Ianos: (i) preceding convection leading to the emergence of a low-PV bubble embedded in a trough, (ii) Ianos's cyclogenesis underneath the southern side of the low-PV bubble, and (iii) strong southerly flow west of the bubble with an anticyclonic curvature north-west of Ianos. The following two sections explore the role of mid-level diabatic processes and upper-level baroclinic forcing on the three features.

### 4.6 Diabatic forcing of the evolution of the upper-tropospheric structure

The diabatic processes in our case are mostly associated with convection occurring during the preceding precipitation event between late on the 14[th] and early on the 15[th] September, as well as during the intensification of Ianos a day later: peaks "A" and "B" in Fig. 5. The impact of diabatic processes on the upper-level features can be explored with the SGT inversion tool, which can determine the balanced circulation induced by the diabatic heating at any particular time of the simulation. The inversion tool integrates the SGT system described in 2.2 with the MetUM dynamical prognostic fields (e.g., winds, potential temperature, humidity) and the diabatic tendencies from MetUM physical parametrizations, summed up in the $S_\theta$ factor (in the second term on the right-hand-side of eq. 1).

To evaluate the ability of the SGT inversion tool to capture the flow and decompose it in diabatic and geostrophically forced parts, we first focus on one event before considering how the diabatic-induced flow evolves throughout all three SST experiments. The event is a strong convective event prior to Ianos's cyclogenesis at $18Z14$ in the simulation with +2 K SST initialized at $00Z14$ and is part of the preceding precipitation event shown in Fig. 4.b. Strong ascent centred at 35°N, 16°E is located inside the low-PV bubble at upper levels, with anticyclonic upper-level winds on its north-western side (Fig. 12.a). The balanced flow from the SGT inversion shows slightly weaker horizontal and vertical winds (Fig. 12.b), and the flow inside the low-PV bubble is more divergent than anticyclonic. These differences can be attributed to the unbalanced flow associated with small-scale processes.

The ageostrophic flow, part of the SGT balanced solution, is markedly divergent around the ascent inside the low-PV bubble (Fig. 13.a), and absent or unorganised elsewhere. Hence, the predominantly anticyclonic upper-level flow surrounding the bubble shown in Fig. 12.b is geostrophic and thus the result of geostrophic adjustment from the temperature gradient between the tropospheric warm air inside the low-PV bubble and the cold stratospheric air outside. There is no ascent nor divergence when the diabatic contribution is neglected in the SGT inversion: the $NO-HEAT$ solution where $S_\theta=0$ (Fig. 13.b). The SGT inverted flow with the diabatic forcing term active and the geostrophic one neglected, the $HEAT-ONLY$ solution, produces a very similar ascent and divergence inside the low-PV bubble to the full solution with both terms (Fig. 13.a,c). Hence, the divergent ageostrophic flow inside the low-PV bubble is mostly driven by the diabatic heating. The divergent flow is ageostrophic by definition, so it is discussed without the ageostrophic adjective hereafter.

The vertical-latitudinal section of the ageostrophic flow averaged between 15.5° and 18.5°E for this convective event preceding Ianos's development is shown in Fig. 14.a. There is strong ascent across the vertical, peaking in the mid levels (∼500 hPa). The convective plume induces convergence at lower levels and divergence at upper levels. The SGT inversion without the

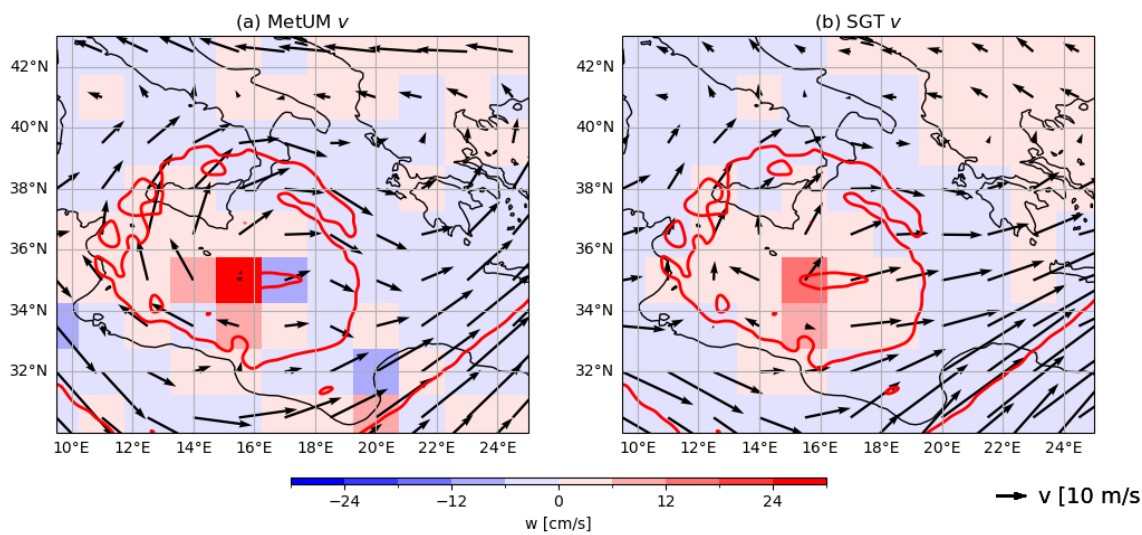

**Figure 12.** Vertical velocity (colours) and horizontal winds (vectors) for the simulation with +2 K SST initialised at $00Z\,14$ at $T+18$, spectrally filtered to $0.32°$ and regridded to $1.5°$, and vertically averaged between model level 54 (at 10.3 km) and 58 (12.6 km) which is roughly equivalent to the 200 and 250 hPa pressure levels. The red contours show the spectrally filtered PV from MetUM at 2 PVU ($< 2$ PVU within contours): (a) MetUM winds, (b) full winds from the SGT inversion tool.

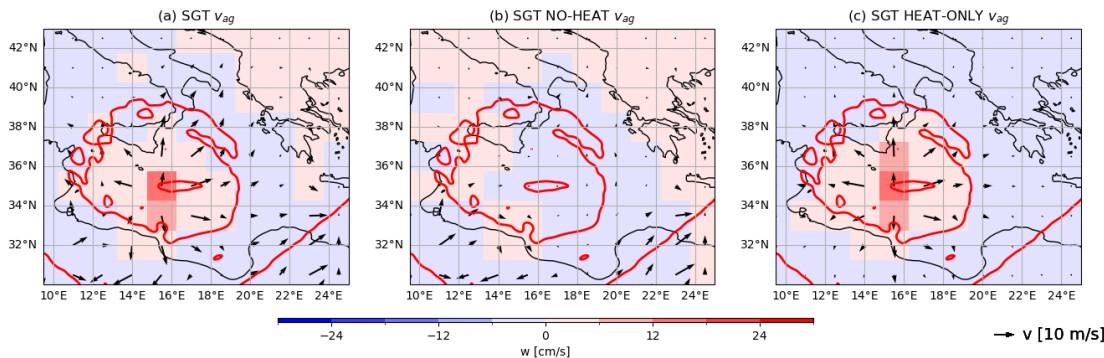

**Figure 13.** Same as Fig. 12 but for the ageostrophic wind inverted from the SGT using (a) full solution with both geostrophic and diabatic forcing terms, (b) SGT inverted flow without heating and (c) SGT inverted flow with only heating. See text for details.

heating contribution shows no ascent nor divergence (Fig. 14.b), and the inversion with only the diabatic forcing active shows

a similar field to the full solution (Fig. 14.a,c). The tropopause elevation on the southward side of the plume is collocated with the divergence contours, highlighting the importance of the diabatic outflow in pushing the tropopause upwards and sidewards. In addition to the event described here, vertical ascent induced by diabatic heating and subsequent production of divergence in the upper levels occurs at other times and simulations. These events are termed "diabatic divergence" or "diabatic outflow" events hereafter.

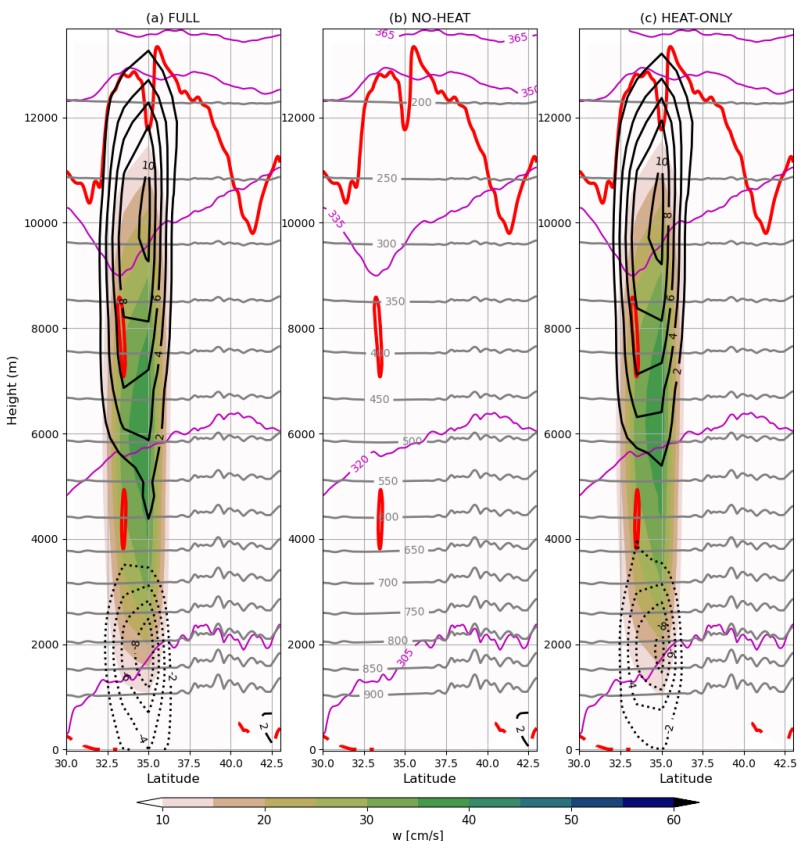

**Figure 14.** Latitudinal cross section for the same event as in Fig. 13 of SGT ageostrophic flow for (a) full solution with both geostrophic and diabatic forcing terms, (b) SGT inverted flow without heating and (c) SGT inverted flow with only heating. Positive values of vertical velocity are shown in colours, divergence in black contours (spaced every $2 \ 10^{-5} \cdot s^{-1}$s from $2 \ 10^{-5} \cdot s^{-1}$) and convergence in black dotted contours (analogous values to divergence but negative). Spectrally filtered to $0.32°$ $\theta$ from MetUM simulations is shown in magenta contours (spaced every 15 K from 305 K), pressure in grey contours (every 50 hPa from 200 hPa) and PV at 2 PVU (red contour). Labels for $\theta$ and pressure are only shown in (b) for clarity.

The time evolution of the diabatic outflow and colocation with the low-PV bubble along the cyclone track is shown in Fig. 15, where the PV and diabatic outflow fields are shown across the points along the three-hourly cyclone track in the $x$-axis, and for each of these fixed points the temporal evolution across forecasts validation times is in the $y$-axis. The dotted diagonal line denotes when the cyclone transits through each location of the track, with the size of dots denoting Ianos's MSLP. Thus at a given $x$-axis value, times below the dotted line are prior to the arrival of the cyclone at the associated location and times above the dotted line are after its arrival. Similarly, at a given $y$-axis value, locations to the left of the line are past track locations and those to the right are future track locations. The locations are defined as the spatial average of a $3° \times 3°$ box with the cyclone location at the centre. The diabatic divergence is vertically averaged between MetUM levels at roughly 250–300 hPa, and PV between 200–250 hPa, both collocated with their maxima in Fig 14.

The intense diabatic outflow at $18Z14$ in the simulation with +2 K SST initialised at time $00Z14$, shown in Figs. 12-14, is the horizontal line at $18Z14$ between the cyclone's locations 17°E, 32°N and 36°E, 16°N in Fig. 15.a. Its flattened appearance on the diagram is the result of the regridding to 1.5° done by the SGT inversion tool, so a single event may stretch over several points where the cyclone transits. Along with the diabatic outflow, the low-PV bubble (white region within the red contour) emerges at the same time as this intense episode over the locations where the cyclone transits from $18Z14$ until $00Z17$. Once the cyclone is underneath the low-PV bubble at $00Z15$, it begins to intensify with frequent diabatic divergence around it, e.g., at $12Z15$, $06Z16$ and $18Z16$–$06Z17$. Neither of the other two simulations initialised at the earliest time ($00Z14$) develop medicane Ianos. There is a short-lived low-PV bubble and weak diabatic outflow after $12Z15$ in the simulation with control SST (Fig. 15.b), but neither of these two elements in the simulation with -2 K SST (Fig. 15.c). The set of simulations initialised at $12Z14$ shows similar differences across SST experiments to the ones initialised 12 hours earlier. The simulation with +2 K SST shows the early emergence of the low-PV bubble and it is co-occuring with the intensification of Ianos, but with weaker diabatic outflow at the time of the first precipitation maxima early on the 15[th] (Fig. 15.d). The simulation with control SST shows weak diabatic divergence between $18Z15$ and $06Z16$ and between 17°E, 32°N and 18°E, 34°N, and a weak cyclone that never deepens beyond 1008 hPa moving in and out of the low-PV bubble (Fig. 15.e). This association suggests that the low-PV bubble influences the early intensification of the cyclone rather than the other way around. Similarly to the previous simulation with -2 K SST, the simulation initialised at $12Z14$ with -2 K SST has no diabatic outflow and no low-PV bubble (Fig. 15.f).

The simulations with +2 K SST initialised at the three latest times also show strong diabatic outflow inside the low-PV bubble (Fig. 15.g,j,m), with the one initialised at $00Z15$ displaying the highest values. However, although this simulation generates stronger diabatic outflow than the earlier two simulations, it does not develop a deeper Ianos (copper line with small dots versus copper lines with pluses and large dots in Fig. 2). Ianos moves ahead of a high-PV area in the two latest simulations before $00Z17$ but in earlier simulations the high-PV area takes longer to reach the same locations, probably due to the strong outflow pushing the high-PV area away from the cyclone, as shown in Figs. 10.a and 11.a,d,g. The first control simulation to develop an intense Ianos, the one initialised at $00Z15$, is the first to show the surface cyclone travelling underneath the low-PV bubble from the 15[th] to 17[th] and associated with areas that have episodic diabatic outflow (Fig. 15.h). Ianos travels ahead of a high-PV area, the PV-streamer south of the low-PV bubble shown in Fig 11.h. The simulations with -2 K SST initialised at the

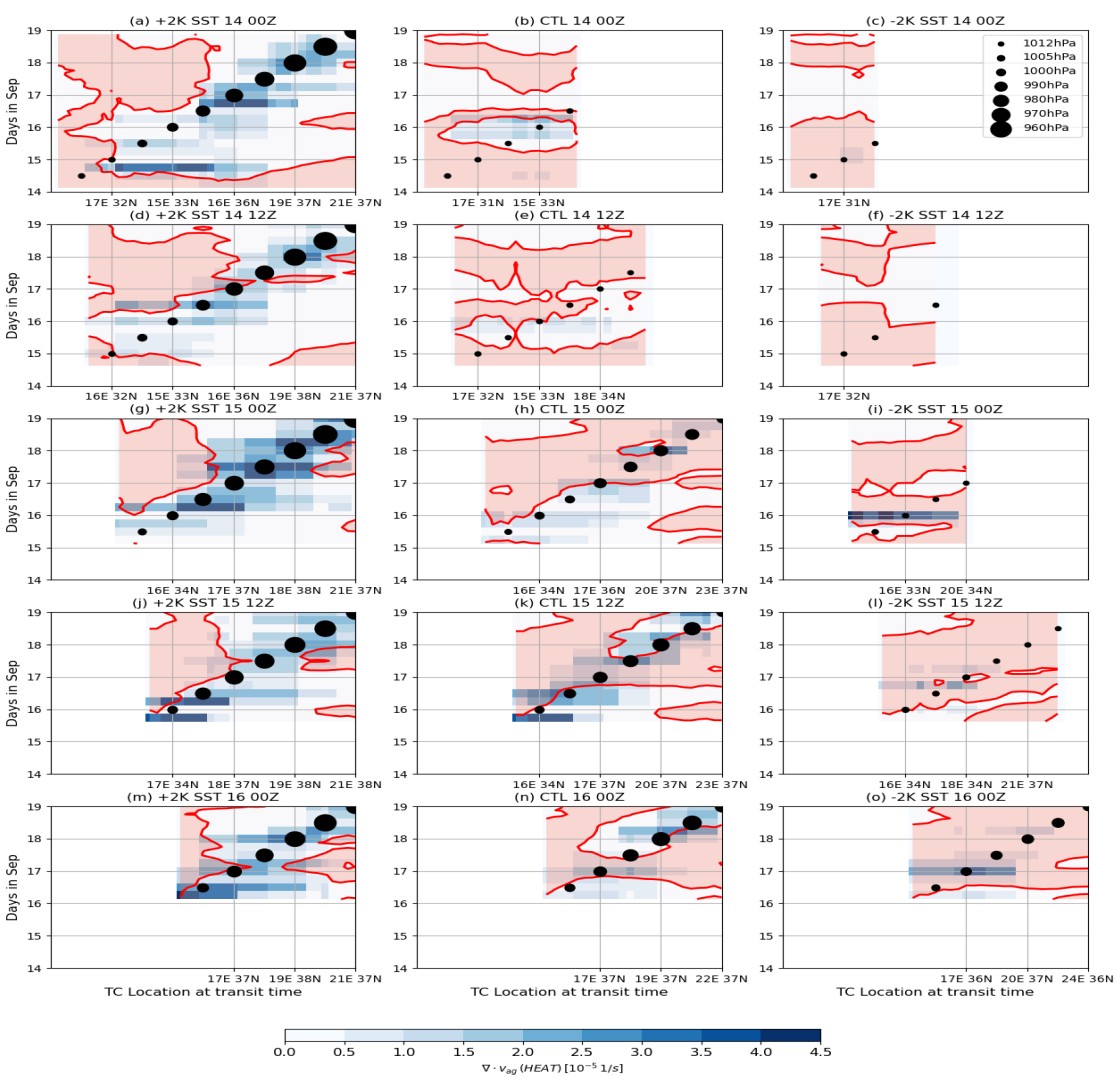

**Figure 15.** Hovmöller of diabatic divergence (colour) vertically averaged between 250–300 hPa, and PV (red shaded for $PV > 2PVU$) between 200–250 hPa. The $y$-axis is validation time of the forecasts (days in September 2020), and each point in the $x$-axis corresponds to a geographical point in Ianos's track. See text for further details. dots on the 1:1 line denote the cyclone depth in MSLP every 12 hours, see legend in (c). Simulations initialised at $00Z\,14$ are (a) +2 K SST, (b) control SST, (c) -2 K SST. Similarly, (d,e,f) for the simulations initialised at $12Z\,14$,(g,h,i) for $00Z\,15$, (j,k,l) for $12Z\,15$ and (m,n,o) for $00Z\,16$. Only points where the cyclone MSLP is below 1012 hPa are shown.

latest three times show episodes of diabatic outflow, but these are weak and isolated in the earliest two, initialised at $00Z\,15$ and $12Z\,15$ (Fig. 15.i,l). In contrast, the one initialised at the latest time shows regular diabatic outflow at $06Z\,16$ and from $18Z\,16$ to $12Z\,17$ (Fig. 15.o) and, unlike the earlier two, the low pressure of Ianos is sustained and the cyclone does not weaken with time (blue line with crosses *vs.* blue lines with diamonds and small dots in Fig. 2).

In summary, the results from the SGT inversion tool suggest that the diabatic heating produced in the preceding precipitation event late on the 14[th] and early on the 15[th] creates a divergent balanced flow in the upper levels, one of the main contributors (if not the only one) to the development of the low-PV bubble. Simulations missing the right timing of the diabatic outflow and low-PV bubble also miss Ianos's intensification. The composites of low-level disturbances near a PV streamer that develop into tropical cyclones detailed in Galarneau et al. (2015) also show a convectively generated divergent outflow located northwest

of a low-level tropical disturbance. The cases where a tropical cyclone does not develop produce a weaker outflow and interact less with the PV streamer. In the Ianos case, the southward side of diabatic outflow generates a PV streamer right above where the low-level pressure perturbation is located (Fig. 8.b for IFS analysis and Fig 10 for simulations). This is an ideal baroclinic environment for Ianos's cyclogenesis. The contribution of upper-level baroclinicity to the early development of Ianos is explored in the next section.

**4.7   Upper-tropospheric baroclinic forcing**

The pair of simulations with control SST initialised at $12Z\,14$ and $00Z\,15$ show a very distinct evolution of Ianos (green line with large dots and diamonds in Fig. 2). However, they both show a low-PV bubble of similar extent at $12Z\,16$ (Fig. 11.e,h). There is weaker connection between the weak surface pressure core, low-PV bubble and diabatic divergence aloft during the 15[th] in the simulation initialised at an earlier time (Fig. 15.e,h), which may lead to a different baroclinic development

of Ianos. Differences in quasi-geostrophic ascent are thus explored with the height-attributable solution from the QG omega equation inversion tool for the simulation with control SST initialised at $12Z\,14$, termed the "poor simulation" hereafter, and one initialised at $00Z\,15$, termed the "good simulation".

The QG ascent forced by all levels is evaluated at the time of Ianos cyclogenesis ($12Z\,15$) at 700 hPa, the same level employed by (Deveson et al., 2002). It is the typical steering level of baroclinic disturbances and also a level with a clear

connection to the near-surface cyclone structure but sufficiently high to avoid boundary layer effects. The good simulation produces a QG ascent peaking at 8 cm/s on the southern area of the low-PV bubble at 16°E, 33°N, which is absent in the bad simulation (Fig. 16). There is clear cyclonic relative vorticity at upper levels above the region of QG ascent and descent in the good simulation (14°E to 16°E and 33°N to 34°N in Fig. 16.a), whereas the upper-level flow is more zonal in the poor simulation.

The longitudinal cross sections at the cyclone latitudes for the QG ascent forced by different level ranges is shown in Fig. 17. In the poor simulation, there is QG ascent on the lower levels below 800 hPa (Fig. 17.a), which is dominated by forcing from the low levels (Fig. 17.b), with much smaller contribution from the mid levels (Fig. 17.c) and marginal contribution from the upper levels (Fig. 17.d). The good simulation, on the other hand, shows a stronger QG ascent than the poor simulation at low levels, and an additional maximum at 650 hPa not present in the poor simulation (Fig. 17.e). QG ascent in this simulation also shows

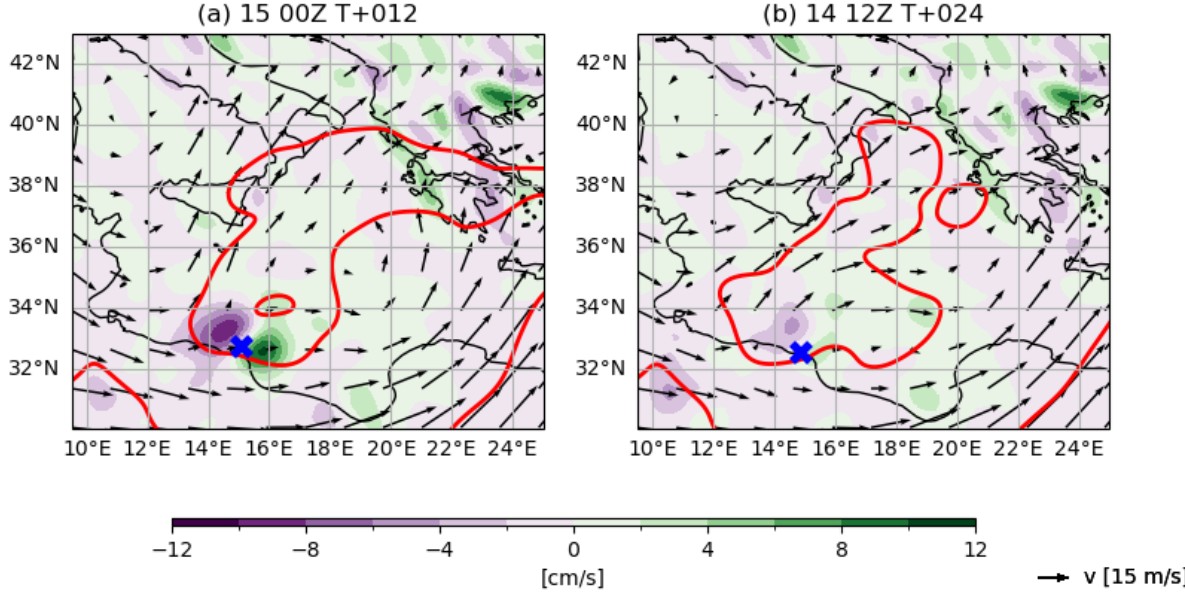

**Figure 16.** QG vertical velocity at 700 hPa and at $12Z\,15$ for simulations with control SST initialised at (a) $00Z\,15$, defined as the "good simulation" in the text, and (b) $12Z\,14$, the "poor simulation". The red contour shows PV at 2 PVU, and vectors show the horizontal wind, both spectrally filtered to $1.6°$ and vertically averaged between 250–200 hPa. The blue 'x' shows the location of the low-level cyclone that later develops into Ianos in the good simulation.

a contribution from mid-level forcing, peaking at 8 cm/s (Fig. 17.g), and from the upper levels peaking, at 4 cm/s (Fig. 17.h). Six hours earlier in the good simulation the contribution from the upper and mid levels to QG ascent is of similar strength with a peak of 4 cm/s, and the contribution from the lower levels halves with a peak of 2 cm/s (not shown). At $12Z\,16$, the time of Ianos's maximum growth rate, the QG ascent peaks at 30 cm/s in the good simulation and is dominated by contributions from low and mid levels (not shown). The QG ascent forced by upper levels at 700 hPa peaks at 1 cm/s in the good simulation at

$12Z\,16$, which is small but comparable to the case of a mid-latitude cyclone at the time of maximum intensification described in Deveson et al. (2002).

Our QG inversion tool does not split the QG ascent into contributions from the VA or TA terms. However, upper-level forcing is generally attributed to the VA term, and the lower-level forcing to TA term. The upper-level strong cyclonic relative vorticity in Fig 16.b, which is advected northeastwards by the background flow, provides a forcing for the VA term. Whereas the strong

horizontal gradients of $\theta$ around the region of QG ascent in the mid levels provides forcing for the TA term (Fig. 17.a,e). For the three validation times evaluated, $06Z\,15$, $12Z\,15$ and $12Z\,16$, the ratio between QG ascent forced by lower and upper levels

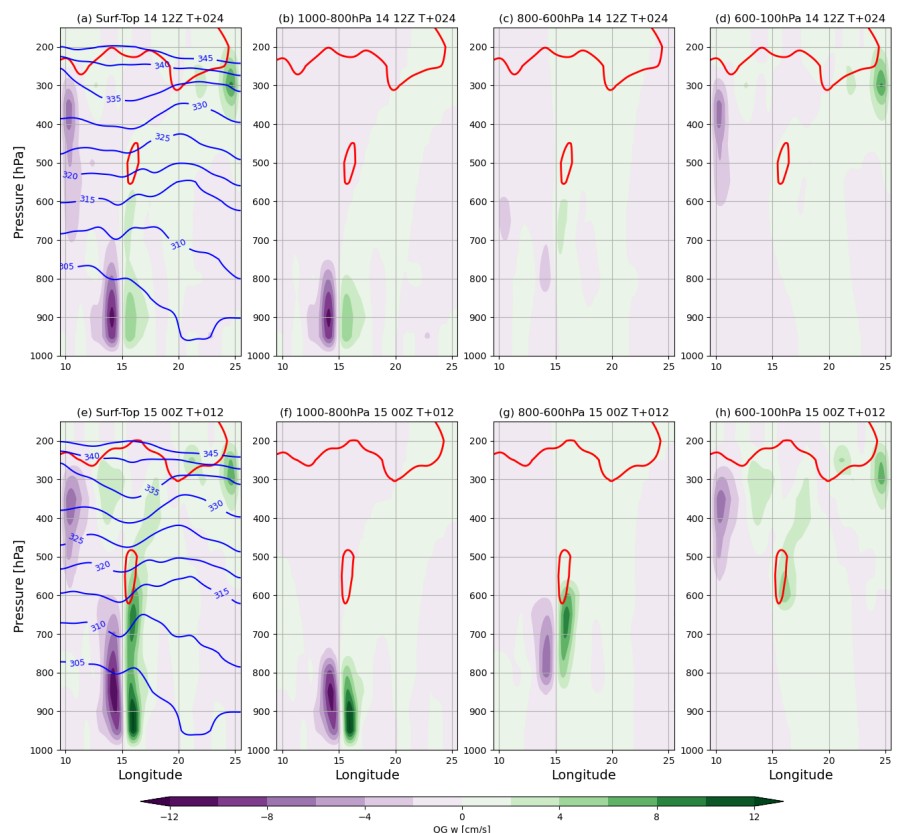

**Figure 17.** Longitudinal cross section of QG vertical velocity (colour) at the cyclone latitude at $12Z\,15$ for (a-d) poor simulation and (e-h) good simulation forced by different level ranges, (a,e) all levels, (b-d) low levels (1000–800 hPa), (c-g) mid-levels (800–600 hPa) and (d,h) upper levels (600–100 hPa). The blue contours show $\theta$ and red contour PV at 2 PVU. Fields are latitudinally averaged between $0.5°$ North and South of the cyclone latitude. The blue 'x' at 950 hPa marks the location of the surface cyclone.

grows with time in the good simulation, such that the relative forcing from lower levels increases with time, whereas in the poor simulation the ratio remains constant, and low levels always dominate over upper levels. In addition, the poor simulation has weaker peaks of QG ascent than the good simulation at all levels. Hence, the preceding convection generates an area of

moist instability at low levels and vorticity at upper-levels. When a weak low with an emerging cyclonic flow at low levels enters this region, it develops a baroclinic cyclone from the contributions from both processes. However, a more zonal flow in the upper levels reduces contribution of the VA term, and thus contributes to the marginal contribution from mid- and lower-level forcing to QG ascent in the poor simulation. The composite of QG ascent in Galarneau et al. (2015) for cases developing tropical transitions is different, the magnitude of their ascent in mid and upper levels is slightly stronger in non-developing

tropical cyclone cases. But they consider cases where the PV streamer is generated by anticyclonic wave breaking instead of local mesoscale convective events.

## 5 Conclusions

Medicanes are a rare and intense type of cyclones that occur in the Mediterranean region. They show tropical-like features such as a cloud-less eye and an axisymmetric warm core, but they are rarely as intense as a full tropical cyclone. Their growth is
driven by a combination of heat fluxes from the sea surface and baroclinic forcing, whose interaction is not yet well-understood. In the present study we investigate the predictability and dynamics of medicane Ianos, one of the most intense medicanes ever recorded, with convection-permitting MetUM simulations at 2.2 km grid spacing. Ianos's cyclogenesis took place on the 15[th] September 2020 over 28°C SST, anomalously 1.5–2 K warmer than climatology, and after intense but unorganised convective precipitation over the area. It became a medicane after developing an axisymmetric warm core and a cloud-less eye early on
the 17[th], making landfall a day later, on the 18[th], over the Ionian Islands, western Greece. Afterwards it moved southwestward, weakened and lost its tropical features on the 19[th]. Our simulations hence start at different initialisation times to capture different processes, including the preceding precipitation and the rapid intensification before Ianos develops an axisymmetric warm core. The SST is homogeneously perturbed from the operational OSTIA analysis by +2 K or -2 K to explore the role of stronger and weaker surface fluxes on Ianos's intensification.

All the simulations with +2 K SST, regardless of the initialisation time, develop a very strong medicane Ianos (some reach a minimum MSLP of 950 hPa). The simulations with control SST develop a medicane only in simulations initialised at or after $00Z15$. There is only one simulation with -2 K SST that develops a medicane, albeit a weak one, which is the one initialised at the latest time ($00Z16$). All of the other simulations with -2 K SST fail to fully develop Ianos. Hence, medicane Ianos might not have existed if SST anomalies in mid-September 2020 were marginal or negative. There is a strong preceding
precipitation event on the 14[th] and early 15[th], ahead of Ianos's cyclogenesis, and over regions where the cyclone later transits and intensifies. This event is only captured in simulations with warmer SST. Note that the improvement of the simulation with warmer SST does not necessarily imply that there is an error in the simulated surface fluxes in the control simulation. Instead, the enhanced fluxes may be compensating for one or more missing or poorly parametrized processes in the MetUM with RAL2M configuration. There is also a second precipitation maxima, occurring around the cyclone at $12Z16$, when it is
intensifying at its maximum rate. The second precipitation peak has been shown to occur in other medicanes (Fita and Flaounas, 2018; Miglietta et al., 2013), but the preceding precipitation event about three days before the peak in medicane intensity has not previously been associated with medicane cyclogensis to the authors' knowledge.

Under a baroclinic framework Mediterranean cyclones intensify from a stratospheric intrusion generating the necessary PV gradients in the upper troposphere (Flaounas et al., 2015, 2021). In the case of Ianos, the paradigm is similar but the origin
of the PV gradients is unusual. The stratospheric intrusion, associated to a homogeneous positive PV anomaly, overshadows the area where Ianos intensifies and, during the preceding precipitation, tropospheric low-PV pockets are deposited aloft that coalesce into a single entity, termed a low-PV bubble here, within the stratospheric intrusion.

The inverted semi-geostrophic flow from the MetUM simulations during the preceding precipitation event shows ascent with divergent outflow in the upper levels that is fully attributable to diabatic heating produced by MetUM physical parametrizations. The divergent outflow pushes the tropopause upwards and sidewards and either drives, or substantially contributes to, the development of the low-PV bubble. The geostrophic vorticity in the upper levels on the 15[th] is thus associated with the low-PV bubble. The advection of this relative vorticity contributes to the baroclinic development of a surface low beneath coming from the Libyan coast. A quasi-geostrophic height-attributable inversion tool is applied to a pair of simulations able and unable to capture Ianos's intensification, and the former shows the presence of QG ascent forced by mid and upper levels. Ianos's diabatic outflow resumes during the 16[th] as it grows and develops a PV tower.

The processes for Ianos's intensification drawn from the analysis of our simulations can be summarised in three steps:

1. The preceding precipitation event deposits or creates low-valued PV in the upper troposphere late on the 14[th] and early on the 15[th], forming a low-PV bubble inside the trough. Diabatic heating released by convective ascent induces a balanced divergent flow in the upper levels that contributes to the low-PV bubble formation and growth.

2. The low-PV bubble is associated with predominantly anticyclonic geostrophic vorticity and a divergent ageostrophic out-flow as it expands early on the 15[th]. The upper-level forcing from the vertical gradient of the advection of the geostrophic vorticity is associated with quasi-geostrophic ascent, which favours Ianos's cyclogenesis on the 15[th].

3. During Ianos's intense growth on the 16[th], a PV tower is formed at its centre and the upper-level flow on its northwest sector turns anticyclonic. There is also divergent diabatic outflow aloft signalling the strength of diabatic processes producing positive PV in the column.

The simulations that follow steps 1 and 2 and develop medicane Ianos either have (a) +2 K SST enabling them to simulate the emergence of the low-PV bubble after the preceding precipitation events or (b) control SST but were initialised at a later time, once the analysis contains the low-PV bubble and associated geostrophic vorticity over the right locations. For the simulations with -2 K SST, only the one initialised at the latest time follows step 3 above, and thus is able to sustain Ianos's intensification. All the other ones are initialised during or before Ianos's cyclogenesis and hence they cannot draw enough energy from surface fluxes to sustain the diabatic activity in the PV tower and the associated divergent diabatic outflow at upper levels.

Ianos's pathway into a medicane shares similarities with the meridional trough category of subtropical cyclones undergoing tropical transition (Bentley et al., 2017). The development of deep convection in this category may be partially induced by QG forcing; these cyclones show a weaker dependency on bulk tropospheric stability and a greater frequency of strong tropical transition events, those able to produce a WISHE mechanism under a pronounced weakening of shear environment from convection upshear (Davis and Bosart, 2003).

The role of preceding or preconditioning precipitation events has been identified here during medicane development for the first time. There is another kind of cyclone transition where preceding precipitation also preconditions the baroclinic environment: the extra-tropical transition of tropical cyclones. During such events, tropical cyclones advect warm and moist air poleward, triggering extratropical cyclogenesis and initiating ridge building and jet acceleration ahead of the tropical cyclone

(Grams and Archambault, 2016), which contributes to the re-intensification of the transitioning (tropical to extra-tropical) cyclone (Keller et al., 2019). In the Ianos case, the preceding precipitation is independent of the developing medicane.

We make the following recommendations for model development from our findings. The representation of convective processes in the preceding precipitation event in the simulations with control SST is poor. Either finer resolutions or better physics such as a scale-aware convection scheme, for example, might improve the simulation of these processes and hence the predictability of cyclones such as medicane Ianos. Additionally, capturing SST evolution is important for medicane prediction, either by coupling to ocean models as done in Ricchi et al. (2017) and Stathopoulos et al. (2020), or updating SST from an operational ocean-only system (Mahmood et al., 2021). The global methodologies to perturb SST in the ensemble system such as the one described in Tennant and Beare (2014), with amplitudes as large as 2 K, may be needed to coarsely represent the uncertainty around medicane intensification.

The simulations with +2 K SST have provided insightful results to interpret the intensification mechanisms, but also a terrifying vision of a 950 hPa cyclone over the Mediterranean (Fig. 7.a). The strength of the medicane in our simulations with +2 SST may be excessive as the near-surface atmospheric temperatures were not adjusted and SSTs were fixed, thus cooling feedbacks between atmosphere and ocean not represented. However, given the relationship between SST and medicane strength found here and elsewhere, e.g. Miglietta et al. (2011) and Noyelle et al. (2019), an accurate representation of SST changes and air-sea interactions in convective-scale regional climate prediction models may be critical to determining the future climatology of medicanes.

*Code availability.* MetUM model is available for use under a closed licence agreement, further information at http://www.metoffice.gov.uk/research/modelling-systems/unified-model (last access: 19 October 2023). The SGT inversion tool is available at the MetOffice Science Repository Service (MOSRS) at https://code.metoffice.gov.uk/svn/utils/sgdiagnostic/ (last access: 19 October 2023)

*Data availability.* MetUM simulations are archived in Met Office Operational Storage Environment (MOOSE) at devfc/u-ci649 (control SST), devfc/u-cl004 (+2K SST) and devfc/u-cl005 (-2K SST), available upon request. IFS analyses are available at the ECMWF's Meteorological Archive Retrieval System (MARS) Catalogue

*Author contributions.* **CS** led the simulations, data analysis, visualization and writing. **SG** provided the Quasi-Geostrophic diagnostics and contributed substantially to conceptualisation and writing. **AV** contributed to conceptualisation and writing. **FP** and **SD** designed the model protocol for the simulations and contributed to writing. **SB** contributed to the conceptualisation and writing.

*Competing interests.* Co-author Silvio Daviolo is a member of the editorial board of WCD.

*Acknowledgements.* This article is based upon work from COST Action CA19109 "MedCyclones", supported by COST (European Cooperation in Science and Technology). Authors would like to thanks Emanuele Gentile, Julian Hemming and John Methven for their scientific and technical advice. We thank two anonymous reviewers for their helpful suggestions to improve the article.

645

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
