# Peer review of "The impact of preceding convection on the development of Medicane Ianos and the sensitivity to sea surface temperature"

_EGUsphere, 2023_

## Author Comment (AC1)

**Response to reviewers of "How a warmer Mediterranean preconditions the upper-level environment for the development of Medicane Ianos "**

We appreciate the time and effort taken by the reviewers to provide a list of insightful comments to improve the paper. A reply to each of their comments is given in blue. Both reviewers coincide in few points, so some replies are cross-referenced. Line and Figure references are for the original manuscript. Figures and equation in the present document and not in the mansucript are refered as "here".

**Anonymous Referee #1**

**General comment**

I find the presentation to be mostly descriptive, despite the use of two quantitative diagnostics. Importantly, I was often left wondering about the significance of described observations. While most observations support the authors' main point about the importance of the upper-level low-PV "bubble" well, the presentation of observations without an evident link to theory or conceptual models make for a rather lengthy read. Personally, I would have much appreciated more such links to conceptual or theoretical frameworks of TC-like storm dynamics. I am aware that this comment is very general, but it would take me an unwarranted amount of time to think more thoroughly about Ianos' evolution to make more specific suggestions for improvement. I consider the presentation "ok" for the purpose of the present study, in the sense that the presentation does provide evidence for the main point of the study. I thus leave it to the authors' discretion to improve their manuscript in this respect.

We acknowledge that our original manuscript did not provide enough detail to understand the results in a tropical transition framework. We will incorporate and describe a figure with the Cyclone Phase Space (CPS) diagram of Hart (2003), with results from the IFS analysis and the simulations. The future new figure is shown in Figure 1 here, it shows that Ianos during the period of early intensification ($16^{th}$) has not attained an axi-symmetric warm-core. Ianos shares similarities with the meridional trough category of subtropical cyclones undergoing Tropical Transitions (Bentley et al., 2017), e.g. quasi-geostrophic forcing and less dependence in bulk stability. We will relate some of our results those found in Tropical Transitions (TT) composites Galarneau et al. (2015), climatologies (McTaggart-Cowan et al., 2008, 2013), metrics (McTaggart-Cowan et al., 2015) and processes (Yanase et al., 2023).

**Major comments**

1. *QG omega analysis:*

   The importance of the low-PV bubble is well supported from a phenomenological perspective and is of interest in itself. The mechanistic link of how these upper-level changes impact Ianos' evolution, however, is not established sufficiently well by the QG omega inversion analysis. I have several issues with the analysis. First, the signal that the authors present is very weak, and the associated discussion is partly confusing (L465: I do not see the absence of vertical motion described by the authors; L475: I am confused if the authors show omega or w. Equation 3 says omega, but here positive values seem to be referred to as ascent.) Only two out of the six simulations are contrasted. It is thus not possible to say if the weak signal is at least systematic. Second, there is no theoretical or conceptual justification of how the observed small differences would impact the evolution of Ianos. Note that arguments based on the dynamics of (fully) extratropical cyclones can be applied here only partially because Ianos has clearly tropical characteristics. Formation and intensification mechanisms of tropical-like storms thus need to be taken into account also. Third, the authors consider one part of the forcing only. Without evidence that the other forcing term is indeed negligible, I consider the analysis to be incomplete. Finally, the authors should acknowledge

[Figure]

Figure 1: Time-series of (a) thickness asymmetry $B$ between 600–925 hPa (b) $V_T^L$ low-level thermal wind between 600–900 hPa and (c) $V_T^L$ upper-level upper thermal wind between 300–600 hPa. Same colours as in Figure 2,3 and 6 of the manuscript. The CPS metrics are computed with a radius of 400km instead of 500km because of the small size of our domain (only points on the $16^{th}$ and $17^{th}$ where captured). The $B$ is computed at 925 hPa level instead of 900 hPa.

the intricacies of a vertical partitioning of forcing in the omega equation, as discussed by Morgan (1999).

(a) We will replace Section 4.6 in the manuscript entirely. the new Figure 15 will show QG ascent forced by the full column. The cross-sections shown in Fig 16 will also include QG ascent forced by the full column and mid-levels.

(b) The methodology section will show equation 3 with the vertical wind component $w$

(c) The introductory paragraph of the new section 4.6 will provide more detail about why only two simulations with control SST are used instead of the full array of simulations

(d) The new section will include a clearer explanation of the attribution of TA and VA terms to the ascent forced by different levels.

(e) We will compare the low values of QG ascent forced by upper levels in Ianos at $12Z$ 16, with those obtained by Deveson et al. (2002) for a mid-latitude cyclone at the time of maximum intensity. The figure for Ianos supporting this statement is not included in the paper but is included here in Fig. 2 here for reference.

[Figure]

Figure 2: Quasi-Geostrophic ascent at 700hPa forced by the upper levels (600-100hPa) for the "good simulation", initialised at 00Z 15, at 12Z 16 validation time. Note values in the colorbar are higher than in Figure 15.

(f) We will update point (ii) in the abstract with the new results. Additionally we will update the description of the section's results in L518 and the second point in the list of key elements at L526.

(g) The partitioning done in Morgan (1999) attributes the vertical motion to individual PV anomalies, using the distribution of the QG forcing from the Q vector convergence. Our methodology is different. We obtain the QG ascent forced by different level ranges from the inversion of the QG equation with the forcing applied at selected levels.

In addition, there is an alternative plausible hypothesis how the changes to the upper levels impact Ianos: by changes to the vertical wind shear. The authors note this mechanism in the discussion of preconditioning role of preceding convection during tropical transition in L542. The same mechanisms may be at work here, too, and analysis of this aspect would provide much benefit to the manuscript.

We consider that the wind-shear hypothesis is weaker than those presented in the manuscript. The Figure 3 here shows the timeseries of vertical wind Shear (VWS), defined as the magnitude of the wind shear vector between the 200 hPa and 850 hPa levels and spatially averaged in a 3x3 box with the cyclone at the centre. The wind shear drops once the surface weak low goes underneath the trough on the $14^{th}$, but it is still above 10m/s in all simulations and the IFS analysis. VWS increases through the cyclone's lifetime in all simulations. The VWS values in those simulations not developing medicane Ianos are not different than those developing it. Very similar results are obtained when averaging VWS with a radius of 150 km around the centre of the cyclone (not shown), as well as a radius around 500 km for IFS-AN (not shown). We will include some comments about wind shear in section 4.1, but without the figure.

[Figure]

Figure 3: Time-series of vertical wind shear around the cyclone, see text above for details.

2. *Title:* I fully agree with the authors that convection preceding Ianos can be considered as a pre-conditioning of subsequent evolution. Warmer SST makes this preconditioning more robust in the model simulations of this specific case. I'd argue, however, that the preconditioning is still by the preceding convection and not by the warmer SST. It is hard to make a general case about the preconditioning role of SST from this single case. I think that it is thus warranted to adjust the title accordingly.
The following title is suggested: The impact of preceding convection in the development of Medi-cane Ianos and the role of sea surface temperature
We would like to keep a reference to the SST, as this paper is the result of the journey we under-took to understand how the predictability of Ianos improved in the simulations with warmer SST experiments. The experiments are very sensitive to SST and the paper's points on the preceding precipitation, low-PV bubble and diabatic heating align well with warmer or cooler SST.

3. *Reference to tropical transition:* At a few points in the manuscript the authors refer to the evolution of Ianos as tropical transition. Is there evidence that Ianos indeed started as a (distinct) midlatitude cyclone? The authors argument in L520 that baroclinic forcing alone was insufficient for strong cyclogenesis speaks against a distinct baroclinic origin of Ianos. An alternative interpretation could consider the observed evolution more akin to tropical cyclogenesis. I believe that clarification or a more explicit discussion of tropical transition vs. tropical cyclogenesis would be beneficial to the manuscript because the underlying mechanism differ distinctly between these pathways.
Yes, Ianos did not have a symmetric warm core during its first day (Figure 1 here). L520 will be rewritten to make clearer that diabatic processes were important during Ianos intensification like in intense mid-latitude cyclones or sub-tropical cyclones.

**Minor comments**

L118: single-moment scheme: Please clarify which hydrometeors are represented. Is there ice? The presentation suggests that there is rain only. More generally, please add a brief discussion on how the simplistic one-moment scheme affects the representation of convection.

Details can be found in the "microphysics" section in RAL2M's reference paper Bush et al. (2023) cited in the manuscript. It includes the snow size distribution parameterization of Field Field et al. (2007), which will be referenced in the new version of the manuscript.

Broadening the point on model deficiencies, we will add description of the main setbacks of the RAL2M configuration, mostly precipitation biases at high and low intensity. Note, RAL2M was an established regional configuration at the time we run our simulations, it was operational in the MetOffice and the National Institute of Water and Atmospheric Research (NIWA) for regional domains over UK and New Zealand.

L281: I do not see an indication of dipole structure in the figure and thus this explanation seems mere speculation. Alternatively, and more simply, the observed tropospheric PV values at upper-levels could simply be due to vertical transport (cross-isentropic as well as adiabatic) of PV from the lower troposphere.
The PV dipoles appeared earlier on when the low-PV bubble is forming at $6Z\,14/9$ over the areas of high convective activity, as shown in Figure 4 here. It is clearer in the MetUM simulations, whose resolution is higher than the IFS-Analysis and in which convection is not parametrized. The low-valued PV could be vertically advected from below and/or generated aloft. We do not aim to resolve such question in the manuscript (e.g. see L523). We will detail that the PV dipoles appear earlier and are not shown in Figure 14.a

[Figure]

Figure 4: Upper-level PV and horizontal winds as Figures 7,9 and 10 but at the 14/9/2020 06Z validation time for (a) IFS-Analysis (b) control SST and (c) +2K SST, both init. at $00Z\,14$.

L303: "horizontal gradients": Did the authors evaluate different contributions to PV or is this speculation? Usually, diabatic heating generates PV anomalies by generating stability anomalies, i.e., changing the vertical gradient of theta.
No, we did not evaluate the vertical or horizontal contribution to the PV production, we will clarify this point in the new revision of the manuscript

Fig 14: I am confused: Is this a Hovmoeller diagram for which the meridional location is adjusted at each time step to match the meridional position of Ianos? I have a hard time understanding this rather unconventional and complex plot and thus suggest further improving the description/ introduction of the format. Or is there a simpler way to depict the relevant information?
Yes, it is a Hovmoeller diagram where the meridional location is the validation time of the simulation, and each point in the zonal is a location of the Ianos' track. We believe the figure's format is often employed to show local changes before and after the passage of a TC, in particular the cold-wake in SST, e.g. see Fig 20 of Thompson et al. (2019). We will rewrite the relevant sentence in the caption and the second sentence of the paragraph introducing the figure at L403.

L526: The authors did not study the adjustment process and thus I recommend not emphasizing this process in the conclusions.
The development of the geostrophic circulation on the western side of the low-PV bubble will not be associated to the geostrophic adjustment.

**Editorial comments**

L10: "bubble" I think it is fine to use this term in the main text (L285) in quotation marks and when supported by a figure. For the purpose of the abstract, I recommend rewording.
The "low-PV bubble" element is quite central in our analysis and in the description of our results. Thus we will improve its definition in the abstract, and use it few times in it to avoid duplication.

L15: The sentence reads as if the divergent outflow aloft sustains Ianos' development. Is this the causality that the authors mean to imply. Or isn't the outflow at this point during the development rather a consequence of Ianos' sustained development? Please clarify.
Point (iii) only applies to Ianos intensification during the $16^{th}$. It is missing in earlier simulations with -2K SST. The whole list is rewritten to improve its readability and separate the three elements with clearer text.

L105: What is the medium-complexity model for surface fluxes? Or do you imply that surface fluxes fully control the diabatic processes analysed with Semi-Geotriptic inversion tool? Please clarify. It is the latter, the text will be rewritten to explicitly relate diabatic processes to surface fluxes and SST changes.

L116: change "resolve" to "represent"
it will be included in the future version of the manuscript

L116: The eye is not a process. Please reword.
"Process" will be replaced by "feature"

L128: The use of the term "trajectory" is not entirely clear to me here.
It means the simulated trajectory of an air parcel by MetUM that is consistent with the maintenance of geostrophic and hydrostatic balance. The text will be rewritten to make this term more explicit

L138: Does this sentence make sense?
No, it is missing the verb! It will be fixed as "The Matrix Q **is** shown ..."

L152 Please note in the following discussion if the SGT inversion is linear.
The SGT inversion is not linear, replacing the wind vector $\vec{u}$ with the continuity equation in equation 1 yields an elliptic equation for the pressure tendency $\partial\phi/\partial t$, shown in eq. 1 here. See sect. 2.3 of Sánchez et al. (2020) for further details. Nevertheless, the residual between the full solution and the sum of both ($NO - HEAT + HEAT - ONLY$") is very small ( 1%). The residual of the case shown in the manuscript's Figure 12 is shown in Figure 5.d here, and given its marginal values we do not consider it is worth including it in Figure 12.

$$\nabla \cdot \mathbf{Q^{-1}}\nabla\left(\frac{\partial\phi}{\partial t}\right) = \nabla \cdot \mathbf{Q^{-1}}\mathbf{H} + \nabla \cdot \vec{u}_g \tag{1}$$

[Figure]

Figure 5: As Figure 12 in the manuscript but including (d), the residual for $w$ (colour) and $\mathbf{v}_{ag}$ (vector). The residual is the difference between the "total" solution (panel a) minus the sum of no heat (panel b) plus the diabatic only (panel c) solutions.

L155: What is your estimate of the Rossby radius of deformation? 0.32° deg would be a rather small estimate. Why do you choose this length scale?
We first filter the MetUM fields to scales equivalent to the global operational models of 0.32°

employed by Sánchez et al. (2020) and Hardy et al. (2023), where the SGT inversion tool was successfully applied. Then a second step is performed where fields are bi-linearly regridded to a grid resolution of 1.5°×1.5°. This procedure allows reasonable computational accuracy in computing scales of 500km or greater (Cullen, 2018). The text will be adapted accordingly.

L208: This statement is not strictly correct because the relation is not monotonic. Please revise.
The text will be rewritten to highlight that the maximum intensity of Ianos is higher in the +2 K SST simulations initialised at the three earliest times, than in the latest two.

L213: "trajectory" (here and elsewhere): "track" is the common term used for tropical-like storms. I suggest revising to avoid confusion.
It will be included in the future version of the manuscript.

L315: Titles of subsections 4.3 and 4.4, these subsections discuss also the tropospheric-deep PV tower not only the upper-level structure.
The section will be rewritten to "Evolution of the upper-level structure and associated PV-tower in the IFS analyses".

L302: I have a hard time to identify the low PV values aloft. Please clarify or change color bar..
Figure 8 will be updated to Fig 6 here, which includes contours for PV=0 in blue, so it is now clearer where the blue areas (showing negative PV) are.

[Figure]

Figure 6: As manuscript's Fig 8 but including blue contours for $PV = 0$ $PUV$

L405: suggest adding: "size of the" dots.
It will be included in the future version of the manuscript.

**Anonymous Referee #2**

**1 General comments**

1. The tittle does not totally represent the content of the study and the conclusions: It focus more on the mechanisms that favor the development of Ianos rather than on the interaction between a warm sea or warmer sea and the convective factors that provoke Ianos cyclogenesis. It deals with this issue, but it is not the main part of the study, in my opinion. Indeed, in the abstract we do not see any focus on this.
See reply to second point in the major comments section of Reviewer #1. We propose to change the title to:
The impact of preceding convection in the development of Medicane Ianos and the role of sea surface temperature

2. The study argues about the importance of baroclinic forcing in the development of Ianos, i.e. from a midlatitude meteorology point of view, but given it is a tropical transition phenomenon, it lacks addressing its development for a tropical meteorology point of view, i.e., the dynamics of convection

and its organization. Therefore, I would explicitly mention this concern about the partial view of this phenomenon in this study when discussing the results and conclusions.

We will include an analysis of the cyclone phase space (CPS) during Ianos and other changes detailed in the reply to the general comment of reviewer #1.

**2 Specific points**

1. I would try to reduce the length of the article, especially Section 4.2, 4.3, 4.4 (which are not the main body of the line of arguments; or also merging 4.3 and 4.4 after summarising) and Conclusions and just focus on the main important aspects for understanding the main message about the mechanisms involved in Ianos' cyclogenesis and the role of the SST (although this is not mainly addressed).

   We will try to make the text more concise in these sections when revising the paper. However, given that, in response to the reviewers' comments we plan to include a new section on cyclone phase space diagnosis, the overall length of the article is not likely to reduce.

2. Section 1 (Introduction):

   (a) L35-40: I would argue that upper-level baroclinic forcing is essential for the development of precursors to medicanes, not for the intensification of medicanes, at least for those robust (tropical) ones.

   The intensification of medicanes is not well understood and could be case dependent. The paragraph containing this line and the following paragraph will be rewritten to make a clearer point about their uncertainties.

3. Sections 4.1 and 4.2:

   (a) L197: What is the argument to say that it becomes a medicane at this time?

   This line will be rewritten in line with the results from the new section on the new cyclone phase space diagnostic. See Figure 1 here for the CPS anlaysis. It will incorporated in the rewritten text.

   (b) Figure 2: As the different initialization times are indicated by a different symbol, I would put just the same color for the CTL, -2K and +2K groups. In this way, the interpretation becomes clearer without too much noise.

   We would prefer the current colour system as it highlights how the simulations with similar SST forcing change across initialisation times more clearly. Moreover, if we kept the same legend for all figures showing timeseries, which reduces space and simplifies the reading, timeseries in Figure 6 would be even harder to differentiate.

   (c) L215-225: It would be nice to make this analysis more complete by adding the CPS or at least the VTU values for each run, to provide a clearer idea of its tropical (or not) structure.

   A new section on the cyclone phase space diagnostic will be added. See also the reply to general comment of reviewer #1

   (d) L233: Could you provide more evidence for the dry-eyes mechanism of triggering these thunderstorms.

   The SEVIRI water vapour image omitted from the manuscript is shown in Fig. 7 here. There is a clear dry filament extending from a low-pressure system situated out in the southern North Atlantic, North of Madeira, crossing Northwesterly through the Northeast section of the Iberian Peninsula, passing south of the Balearic islands, bending above North Africa and ending on the Sidra Gulf where convection is taking place South of Sicily. This image is publicly accessible (after registration) from the referenced CEDA data centre.

   Further work could be done to attribute the event to the dry-eye mechanism, e.g. check whether the filament is collocated with a stratospheric PV filament. However, this attribution exercise is described with a speculative tone in our manuscript ("one possible mechanism ..."). We chose to avoid showing Fig. 7 and undertaking the attribution work to reduce the length of the paper. Detailing this process is beyond the scope of the paper..

4. Section 4.3:

   (a) L304-305: Why the authors compare this development with those of extratropical cyclones and not with those of tropical transition? I would argue that this is a typical development for tropical transition cyclones and the result of Ianos acquiring a more robust tropical structure.

[Figure]

EWXT11 MSG 6.2µm WV 13/09/2020 1200 UTC

Figure 7: SEVIRI water vapour image at $12Z$ 13, see manuscript for further details.

Panel (c) in Fig. 8 shows the $12Z$ 16 validation time, when Ianos has not yet developed an axisymmetric warm core (Figure 1 here). Panel (d) shows Ianos where it has already developed the axi-symmetric warm core. We will include a comparison with the results of Galarneau et al. (2015), where the PV tower of surface lows near PV streamers is compared between cases undergoing TT with those which do not.

(b) L307: Talking about tropical-like transition here is not suitable as this is not a rubust definition in the literature. The best way to deal with this type of development is talking about the tropical transition process, to be more in line with the community that study cyclone transitions.
The term "tropical-like transition" will be replaced by medicane or axi-symmetric warm core where applicable. We will detail the process of tropical-like transition for medicanes in the introduction in the revised paper.

(c) In this section and the rest of the sections, I miss discussions about the upper-level dynamics that considers the jet streak behaviour. It is true that the PV field is related to the jet dynamics, but it would be interesting to also consider the analysis/explanations from the jet's QG forcing perspective in the analysis of this section and the following ones. There is not a clear relationship between the position of the surface low and upper-level jets at the time of Ianos cyclogenesis (Fig. 8.b). Ianos sits at the southern tip of the southerly flow while it intensifies one day later (Fig 8.c) as well as near the left jet exit region of the southwesterly jet streak. Hence, Ianos is in an optimal location for intensification relative to the southwesterly jet streak but the interpretation considering both nearby jets is not straightforward.

5. Sections 4.5 and 4.6:

(a) MetUM means full solution?
MetUM means that the $\theta$, PV and pressure fields shown are from the spectrally filtered simulation output (as detailed in caption of Fig 12). These fields are shown mostly for indicative purposes associated to relevant dynamical processes. It will be clarified in the caption

(b) L403: Is not the y-axis showing the dates? (Figure 14).
The description of Figure 14 in text and caption will been changed, see reply to "(Fig. 14)" minor comment from reviewer #1

(c) L467 By watching Figure 16 (a-b), one could argue that the forcing at lower levels is stronger in (a) and, therefore, conclusions derived from Figure 15 are subject to the level of choice (700 hPa). If we choose 800 hPa or 850 hPa, for instance, conclusions could be different. Indeed,

[Figure]

Figure 8: Same as Figure 7 in the manuscript but showing wind speed contours instead of wind vectors, every 10 m/s.

the signal seems to decrease a lot at the 700 hPa level. Please, elaborate more on this. Why not choose 800 hPa (it would still avoid boundary layer effects)? This argument of similar low-level forcing appears to be quite weak.

It is the level chosen by Deveson et al. (2002). See reply to the first major comment of reviewer #1. The text and figures in the section on quasi-geostrophic analysis will be edited in the revised version.

6. Section 6 (conclusions):

(a) L538-544: I see the idea of the authors here, but I don't fully see the relationship between Ianos' development and this mechanism in extratropical transitions and downstream cascade of events. The link is about the upper-level PV modification by a diabatic source, but just up to this point.

Although we are considering a medicane rather than a tropical cyclone here, the extratropical transition of a TC is relevant as it is an example of diabatic processes preconditioning environmental baroclinicity and cyclone reinforcement. Thus we think it is relevant to highlight this evolution for those readers interested in how diabatic processes and baroclinicity interact with each other. The sentences will be rewritten to make this point clearer

(b) L541-544: An analysis focusing on the tropical transition perspective would make the article more complete. The paragraph will be replaced by one which details the TT pathway from meridional thought precursors from Bentley et al. (2017), with further references to TT metrics and processes.

(c) 545-552 I think this part is not very well connected with Ianos' mechanisms and it could derive some confusion.

These lines will be removed and replaced by a paragraph liking our results to the TT literature detailed in the reply above.

(d) L562: I would not mention this hypothetical situation as (as you discuss later) the model is uncoupled and a robust analysis about model performance in this case has not been undergone. This scenario could be quite fictitious.

We think this paragraph provides enough context to understand how "fictitious" our simulations with +2K SST could be. Its purpose is mainly to motivate future work to understand the role of SST under a changing climate and its possible impact on medicane development, as well as to advocate for the use of regional coupled models at convective scales for climate

studies. The end of the last sentence of the paragraph will be rewritten to refocus its point into a modelling perspective, where our results are more relevant.

**References**

A. M. Bentley, L. F. Bosart, and D. Keyser. Upper-tropospheric precursors to the formation of subtropical cyclones that undergo tropical transition in the north atlantic basin. *Monthly Weather Review*, 145(2): 503 – 520, 2017. doi: 10.1175/MWR-D-16-0263.1.

M. Bush, I. Boutle, J. Edwards, A. Finnenkoetter, C. Franklin, K. Hanley, A. Jayakumar, H. Lewis, A. Lock, M. Mittermaier, S. Mohandas, R. North, A. Porson, B. Roux, S. Webster, and M. Weeks. The second met office unified model–jules regional atmosphere and land configuration, ral2. *Geosci. Model Dev.*, 16(6):1713–1734, 2023. doi: 10.5194/gmd-16-1713-2023.

M. Cullen. The use of semigeostrophic theory to diagnose the behaviour of an atmospheric gcm. *Fluids*, 3(4), 2018. ISSN 2311-5521. doi: 10.3390/fluids3040072.

A. C. L. Deveson, K. A. Browning, and T. D. Hewson. A classification of FASTEX cyclones using a height-attributable quasi-geostrophic vertical-motion diagnostic. *QJRMS*, 128(579):93–117, Jan. 2002. doi: 10.1256/00359000260498806.

P. R. Field, A. J. Heymsfield, and A. Bansemer. Snow size distribution parameterization for midlatitude and tropical ice clouds. *Journal of the Atmospheric Sciences*, 64(12):4346 – 4365, 2007. doi: 10.1175/2007JAS2344.1.

T. J. Galarneau, R. McTaggart-Cowan, L. F. Bosart, and C. A. Davis. Development of north atlantic tropical disturbances near upper-level potential vorticity streamers. *Journal of the Atmospheric Sciences*, 72(2):572 – 597, 2015. doi: 10.1175/JAS-D-14-0106.1.

S. Hardy, J. Methven, J. Schwendike, B. Harvey, and M. Cullen. Examining the dynamics of a borneo vortex using a balance approximation tool. *EGUsphere*, 2023:1–31, 2023. doi: 10.5194/egusphere-2023-1312.

R. E. Hart. A cyclone phase space derived from thermal wind and thermal asymmetry. *mwr*, 131(4):585 – 616, 2003. doi: 10.1175/1520-0493(2003)131¡0585:ACPSDF¿2.0.CO;2.

R. McTaggart-Cowan, G. D. Deane, L. F. Bosart, C. A. Davis, and T. J. Galarneau. Climatology of tropical cyclogenesis in the north atlantic (1948–2004). *Monthly Weather Review*, 136(4):1284 – 1304, 2008. doi: 10.1175/2007MWR2245.1.

R. McTaggart-Cowan, T. J. Galarneau, L. F. Bosart, R. W. Moore, and O. Martius. A global climatology of baroclinically influenced tropical cyclogenesis. *Monthly Weather Review*, 141(6):1963 – 1989, 2013. doi: 10.1175/MWR-D-12-00186.1.

R. McTaggart-Cowan, E. L. Davies, J. G. Fairman, T. J. Galarneau, and D. M. Schultz. Revisiting the 26.5°c sea surface temperature threshold for tropical cyclone development. *Bulletin of the American Meteorological Society*, 96(11):1929 – 1943, 2015. doi: 10.1175/BAMS-D-13-00254.1.

M. C. Morgan. Using piecewise potential vorticity inversion to diagnose frontogenesis. part i: A partitioning of the q vector applied to diagnosing surface frontogenesis and vertical motion. *Monthly Weather Review*, 127(12):2796 – 2821, 1999. doi: 10.1175/1520-0493(1999)127¡2796:UPPVIT¿2.0.CO;2.

C. Sánchez, J. Methven, S. Gray, and M. Cullen. Linking rapid forecast error growth to diabatic processes. *qjrms*, 146(732):3548–3569, 2020. doi: 10.1002/qj.3861.

B. Thompson, C. Sanchez, X. Sun, G. Song, J. Liu, X.-Y. Huang, and P. Tkalich. A high-resolution atmosphere–ocean coupled model for the western maritime continent: development and preliminary assessment. *Climate Dynamics*, 52:3951–3981, 2019. doi: 10.1007/s00382-018-4367-0.

W. Yanase, U. Shimada, N. Kitabatake, and E. Tochimoto. Tropical transition of tropical storm kirogi (2012) over the western north pacific: Synoptic analysis and mesoscale simulation. *Monthly Weather Review*, 151(10):2549 – 2572, 2023. doi: 10.1175/MWR-D-22-0190.1.

---

## Author Response (AR1)

**point-by-point response to all referee comments for**
**egusphere-2023-2431**

We appreciate the time and effort taken by the reviewers to provide a list of insightful comments to improve the manuscript *egusphere-2023-2431*, originally titled "How a warmer Mediterranean preconditions the upper-level environment for the development of Medicane Ianos". A reply to the reviewers comments is given in blue and changes to the manuscript are reported here in red. Line numbers refer to the new revision of the manuscript, unless explicitly stated. Same for figure numbers. Figures and equations in the present document are refereed as "here". Some replies refer to the replies made in the author response to the interactive comments, whereas others are copied here.

**Anonymous Referee #1**

**General comment**

I find the presentation to be mostly descriptive, despite the use of two quantitative diagnostics. Importantly, I was often left wondering about the significance of described observations. While most observations support the authors' main point about the importance of the upper-level low-PV "bubble" well, the presentation of observations without an evident link to theory or conceptual models make for a rather lengthy read. Personally, I would have much appreciated more such links to conceptual or theoretical frameworks of TC-like storm dynamics. I am aware that this comment is very general, but it would take me an unwarranted amount of time to think more thoroughly about Ianos' evolution to make more specific suggestions for improvement. I consider the presentation "ok" for the purpose of the present study, in the sense that the presentation does provide evidence for the main point of the study. I thus leave it to the authors' discretion to improve their manuscript in this respect.

We acknowledge that the original manuscript did not provide enough detail to understand the results in a tropical transition framework. We have incorporated a figure with the Cyclone Phase Space (CPS) diagram of Hart (2003)—this new figure shows that Ianos has not attained an axisymmetric warm-core during the period of early intensification ($16^{th}$). Ianos shares similarities with the meridional trough category of subtropical cyclones undergoing Tropical Transitions (Bentley et al., 2017), e.g. quasi-geostrophic forcing and less dependence in bulk stability. We relate some of our results to those found in the Tropical Transitions (TT) composites of Galarneau et al. (2015), TT climatologies (McTaggart-Cowan et al., 2008, 2013), metrics (McTaggart-Cowan et al., 2015), and processes (Yanase et al., 2023).

We have added the following changes to the manuscript to address this point:

1. Introduction: We add more detail about the relation of TT and Medicanes, see reply to reviewer #2 specific point 2.a.

2. CPS results: We include an analysis with a Figure. See reply to reviewer #2 specific point 3.c

3. Comparison of our results with the composites of Galarneau et al. (2015) for cyclones developing or not developing a TT.

   (a) Comparison of the PV tower between cases undergoing TT and those that do not, described in the reply to reviewer #2 specific point 4.a

   (b) Comparison of the convective outflow at L499, copied below
   The composites of low-level disturbances near a PV streamer that develop into tropical cyclones detailed in Galarneau et al. (2015) also show a convectively generated divergent outflow located northwest of a low-level tropical disturbance. The cases where a tropical cyclone does not develop produce a weaker outflow and interact less with the PV streamer.

   (c) Comparison of the QG ascent at L544, copied below
   The composite of QG ascent in Galarneau et al. (2015) for cases developing tropical transitions

is different, the magnitude of their ascent in mid and upper levels is slightly stronger in non-developing tropical cyclone cases. But they consider cases where the PV streamer is generated by anticyclonic wave breaking instead of local mesoscale convective events.

4. The anticyclonic wrapping of the PV-streamer around the cyclone is explained using the phases of Yanase et al. (2023) at L364. copied below.
Yanase et al. (2023) describes the cyclone wrap up for the tropical transition of tropical cyclone Kirogi as an intermediate phase between a baroclinic phase, with a southward upper-level cold trough and a northward low-level warm moist air, and a convective phase with a developed deep warm-core convective vortex.

5. Conclusions: Our findings are associated to the meridional trough category of precursors of sub-tropical cyclones undergoing tropical transition of Bentley et al. (2017): See reply to reviewer #2 specific comments 6.a and 6.b.

**Major comments**

1. *QG omega analysis:* The importance of the low-PV bubble is well supported from a phenomenological perspective and is of interest in itself. The mechanistic link of how these upper-level changes impact Ianos' evolution, however, is not established sufficiently well by the QG omega inversion analysis. I have several issues with the analysis. First, the signal that the authors present is very weak, and the associated discussion is partly confusing (L465: I do not see the absence of vertical motion described by the authors; L475: I am confused if the authors show omega or w. Equation 3 says omega, but here positive values seem to be referred to as ascent.) Only two out of the six simulations are contrasted. It is thus not possible to say if the weak signal is at least systematic. Second, there is no theoretical or conceptual justification of how the observed small differences would impact the evolution of Ianos. Note that arguments based on the dynamics of (fully) extratropical cyclones can be applied here only partially because Ianos has clearly tropical characteristics. Formation and intensification mechanisms of tropical-like storms thus need to be taken into account also. Third, the authors consider one part of the forcing only. Without evidence that the other forcing term is indeed negligible, I consider the analysis to be incomplete. Finally, the authors should acknowledge the intricacies of a vertical partitioning of forcing in the omega equation, as discussed by Morgan (1999).

   (a) We have replaced Section 4.6 in the manuscript entirely (see the new section at L506). It also replaces previous Fig 15 (Fig 16 in the new version of the manuscript) with a figure showing the QG ascent forced by the full column. The cross-sections shown in former Fig 16 (now Fig 17) is also replaced by a Figure showing QG ascent forced by the full column and mid-levels.

   (b) The methodology section now shows equation 3 with the vertical wind component $w$ as in eq. 1 here. The following changes have been made to the text in section 2.3:

   $$N^2\nabla_H^2 w + f^2\frac{\partial^2 w}{\partial^2 z} = f\frac{\partial}{\partial z}\big[v_g \cdot \nabla\zeta_g\big] - \frac{g}{\theta_0}\nabla_H^2\big[v_g \cdot \nabla\theta\big] \tag{1}$$

   L196: A conventional formulation using height as the vertical coordinate (such that it is an expression for vertical velocity, w, rather than omega)

   200: Where $N$ is the buoyancy frequency, $\nabla_H$ the horizontal gradient and $\zeta_g$ is the geostrophic relative vorticity

   (c) The introductory paragraph of the new section at L506, copied below, gives more detail on the decision to show only two simulations, instead of the full array of simulations.
   The pair of simulations with control SST initialised at $12Z\,14$ and $00Z\,15$ show a very distinct evolution of Ianos (green line with large dots and diamonds in Fig. 2). However, they both show a low-PV bubble of similar extent at $12Z\,16$ (Fig. 11.e,h). There is weaker coupling between the weak surface pressure core, low-PV bubble and diabatic divergence aloft during the $15^{\text{th}}$ in the simulation initialised at an earlier time (Fig. 15.e,h), which may lead to a different baroclinic development of Ianos. Differences in quasi-geostrophic ascent are thus explored with the height-attributable solution from the QG omega equation inversion tool for the simulation with control SST initialised at $12Z\,14$, termed the "poor simulation" hereafter, and one initialised at $00Z\,15$, termed the "good simulation".

(d) The new section includes a clearer explanation of the attribution of TA and VA terms to the ascent forced by different levels at L533, copied below.
Our QG inversion tool does not split the QG ascent into contributions from the VA or TA terms. However, upper-level forcing is generally attributed to the VA term, and the lower-level forcing to TA term.

(e) The low values of QG ascent forced by upper levels in Ianos at $12Z\,16$ are compared with those obtained by Deveson et al. (2002) for a mid-latitude cyclone at the time of maximum intensity. The comparison is at L530, copied below.
The QG ascent forced by upper levels at 700 hPa peaks at 1 cm/s in the good simulation at $12Z\,16$, which is small but comparable to the case of a mid-latitude cyclone at the time of maximum intensification described in Deveson et al. (2002).

(f) We have updated the second point of the necessary conditions for the development of medicane Ianos in the abstract at L15, copied below.
a quasi-geostrophically ascent forced by mid and upper levels during Ianos's cyclogenesis, partially associated with the geostrophic vorticity advection which is enhanced by the growth and advection of the low-PV bubble

(g) We have updated the description of the section's results in the conclusions at L582.
The geostrophic vorticity in the upper levels on the $15^{\text{th}}$ is thus associated with the low-PV bubble. The advection of this relative vorticity contributes to the baroclinic development of a surface low beneath coming from the Libyan coast. A quasi-geostrophic height-attributable inversion tool is applied to a pair of simulations able and unable to capture Ianos's intensification, and the former shows the presence of QG ascent forced by mid and upper levels.

(h) We have updated the second point in the list of key elements of medicane Ianos development in the conclusions at L591, copied below.
The low-PV bubble is associated with predominantly anticyclonic geostrophic vorticity and a divergent ageostrophic outflow as it expands early on the $15^{\text{th}}$. The upper-level forcing from the vertical gradient of the advection of the geostrophic vorticity is associated with quasi-geostrophic ascent, which favours Ianos's cyclogenesis on the $15^{\text{th}}$.

(i) The partitioning done in Morgan (1999) attributes the vertical motion to individual PV anomalies, using the distribution of the QG forcing from the Q vector convergence. Our methodology is different. We obtain the QG ascent forced by different level ranges from the inversion of the QG equation with the forcing applied at selected levels.

In addition, there is an alternative plausible hypothesis how the changes to the upper levels impact Ianos: by changes to the vertical wind shear. The authors note this mechanism in the discussion of preconditioning role of preceding convection during tropical transition in L542. The same mechanisms may be at work here, too, and analysis of this aspect would provide much benefit to the manuscript.

We have added the paragraph at L300 (copied below) to indicate that the wind-shear hypothesis is weaker than the QG ascent presented in the manuscript. For further details see the relevant reply in the authors response to interactive comments.
Wind shear at the time of cyclogenesis is between 10 to 15 m/s in the IFS analysis and all simulations, and increases uniformly thereafter (not shown), even in the simulations not developing medicane Ianos.

2. *Title:* I fully agree with the authors that convection preceding Ianos can be considered as a preconditioning of subsequent evolution. Warmer SST makes this preconditioning more robust in the model simulations of this specific case. I'd argue, however, that the preconditioning is still by the preceding convection and not by the warmer SST. It is hard to make a general case about the preconditioning role of SST from this single case. I think that it is thus warranted to adjust the title accordingly.
See new title below. We keep a reference to the SST, as this paper is the result of the journey we undertook to understand why Ianos was captured in simulations with warmer SST and initialised on the $14^{\text{th}}$. Our simulations are very sensitive to SST, as are the paper's points on the preceding precipitation, low-PV bubble and diabatic heating. They all decrease or decrease with warmer or cooler SST.
The impact of preceding convection on the development of Medicane Ianos and the sensitivity to sea surface temperature

3. *Reference to tropical transition:* At a few points in the manuscript the authors refer to the evolution of Ianos as tropical transition. Is there evidence that Ianos indeed started as a (distinct) midlatitude cyclone? The authors argument in L520 that baroclinic forcing alone was insufficient for strong cyclogenesis speaks against a distinct baroclinic origin of Ianos. An alternative interpretation could consider the observed evolution more akin to tropical cyclogenesis. I believe that clarification or a more explicit discussion of tropical transition vs. tropical cyclogenesis would be beneficial to the manuscript because the underlying mechanism differ distinctly between these pathways.

Yes, Ianos did not have a symmetric warm core during its first day (Figure 6). The former L520 has been rewritten (now at L587 and copied below). It now avoids the confusion between diabatic processes and genesis.

The processes for Ianos's intensification drawn from the analysis of our simulations can be summarised in three steps:

**Minor comments**

L118: single-moment scheme: Please clarify which hydrometeors are represented. Is there ice? The presentation suggests that there is rain only. More generally, please add a brief discussion on how the simplistic one-moment scheme affects the representation of convection.

Details can be found in the "microphysics" section in RAL2M's reference paper Bush et al. (2023) cited in the manuscript. It includes the snow size distribution parameterization of Field (Field et al., 2007). Added to L135, copied below

and the snow size distribution parameterization of Field et al. (2007).

Broadening the point on model deficiencies, we have added description of the main setbacks of the RAL2M configuration at L138, copied below.

Despite its operational use in regional models over the UK since December 2019, the RAL2M configuration suffers from heavy precipitation biases, producing too much intense rainfall and too little drizzle (Bush et al., 2023). The use of the two momentum Cloud AeroSol Interacting Microphysics (CASIM Field et al., 2023, ;) reduces some of this bias and will be included in future RAL configurations along with other improvements (Bush et al., 2024).

L281: I do not see an indication of dipole structure in the figure and thus this explanation seems mere speculation. Alternatively, and more simply, the observed tropospheric PV values at upper-levels could simply be due to vertical transport (cross-isentropic as well as adiabatic) of PV from the lower troposphere.

The PV dipoles appeared earlier on when the low-PV bubble is forming at $6Z\,14/9$ over the areas of high convective activity, as shown in Fig. 1 here. MetUM simulations, whose resolution is higher than the IFS analysis and in which convection is not parametrized has clear dipoles in the Gulf of Sidra. The low-valued PV could be vertically advected from below and/or generated aloft. We do not aim to resolve this question in the manuscript (e.g. see L588). We have rewritten these lines at L328, copied below.

Although distinct PV dipoles are not evident in Fig. 8.a, small-scale PV dipoles appear in the upper levels at the earlier stages of convection in the IFS analysis and MetUM simulations (not shown).

[Figure]

Figure 1: Upper-level PV and horizontal winds as Figures 8,10 and 11 but at the 14/9/2020 06Z validation time for (a) IFS-Analysis (b) control SST and (c) +2K SST, both init. at $00Z\,14$.

L303: "horizontal gradients": Did the authors evaluate different contributions to PV or is this speculation? Usually, diabatic heating generates PV anomalies by generating stability anomalies, i.e., changing the vertical gradient of theta.

No, we did not evaluate the vertical or horizontal contribution to the PV production, we replace this point at L351, copied below.

Steady convective ascent along the column generates diabatic heating, e.g. via latent heat release, and thus generates positive PV anomalies that extend along the column as shown in Fig 4 of Wernli and Davies (1997), the so-called PV-tower.

Fig 14: I am confused: Is this a Hovmoeller diagram for which the meridional location is adjusted at each time step to match the meridional position of Ianos? I have a hard time understanding this rather unconventional and complex plot and thus suggest further improving the description/ introduction of the format. Or is there a simpler way to depict the relevant information?

Yes, it is a Hovmoeller diagram where the meridional location is the validation time of the simulation, and each point in the zonal is a location of the Ianos' track. We believe the figure's format is often employed to show local changes before and after the passage of a TC, in particular the cold-wake in SST, e.g. see Fig 20 of Thompson et al. (2019). We have rewritten the relevant sentence in the caption as

The $y$-axis is validation time of the forecasts (days in September 2020), and each point in the $x$-axis corresponds to a geographical point in Ianos's track.

The description of the figure in the text has been rewritten at L458, copied below.

where the PV and diabatic outflow fields are shown across the points along the three-hourly cyclone track in the $x$-axis, and for each of these fixed points the temporal evolution across forecasts validation times is in the $y$-axis

L526: The authors did not study the adjustment process and thus I recommend not emphasizing this process in the conclusions.

The attribution of the geostrophic circulation on the western side of the low-PV bubble to geostrophic adjustment has been removed. The entire sentence has been replaced as described in the reply to reviewer's #1 major comment 1.h

**Editorial comments**

L10: "bubble" I think it is fine to use this term in the main text (L285) in quotation marks and when supported by a figure. For the purpose of the abstract, I recommend rewording.

The "low-PV bubble" element is quite central in our analysis and in the description of our results. Thus we have improve its definition in the abstract at L12 as

An area of low-valued potential vorticity (PV), termed a "low-PV bubble"

L15: The sentence reads as if the divergent outflow aloft sustains Ianos' development. Is this the causality that the authors mean to imply. Or isn't the outflow at this point during the development rather a consequence of Ianos' sustained development? Please clarify.

Point (iii) only applies to Ianos's intensification during the $16^{th}$. It is missing in earlier simulations with -2K SST. The list has been rewritten at L12-18, copied below, to improve its readability and separate the three elements with clearer text.

First, an area of low-valued potential vorticity (PV), termed a "low-PV bubble", formed within a trough above where Ianos developed; diabatic heating associated with a preceding precipitation event triggered a balanced divergent flow in the upper-levels which contributed to the creation and maintenance of this low-PV bubble as shown by results from a semi-geostrophic inversion tool. Second, a quasi-geostrophically ascent was forced by mid and upper levels during Ianos's cyclogenesis, it is partially associated with the geostrophic vorticity advection, which is enhanced by the growth and advection of the low-PV bubble. Third, diabatic heating dominated by deep convection formed a vertical PV tower during Ianos's intensification and continued to produce diabatically-induced divergent outflow aloft, thus sustaining Ianos's development.

L105: What is the medium-complexity model for surface fluxes? Or do you imply that surface fluxes fully control the diabatic processes analysed with Semi-Geotriptic inversion tool? Please clarify.

Yes, the text has been rewritten at L118, copied below, to explicitly relate diabatic processes to surface fluxes and SST changes.

Two medium complexity models are employed to understand the importance of diabatic processes,

enhanced by surface fluxes induced by warmer SST or suppressed by cooler SST, and upper-level baroclinicity in Ianos's intensification.

L116: change "resolve" to "represent"
Done at L130.

L116: The eye is not a process. Please reword.
"Process" has been replaced by "feature" at L130.

L128: The use of the term "trajectory" is not entirely clear to me here.
It means the simulated trajectory of an air parcel by MetUM that is consistent with the maintenance of geostrophic and hydrostatic balance. The text has been rewritten at L146, copied below, to make this term more explicit.
The inversion tool is a single timestep integration of the SG system with atmospheric fields produced by a MetUM simulation. Hence, it provides a solution for the MetUM flow consistent with the maintenance of geostrophic and hydrostatic balance

L138: Does this sentence make sense?
No, it is missing the verb! It is fixed at L157.
The Matrix Q is shown

L152 Please note in the following discussion if the SGT inversion is linear.
The SGT inversion is not linear, replacing the wind vector $\vec{u}$ with the continuity equation in equation 1 yields an elliptic equation for the pressure tendency $\partial\phi/\partial t$, shown in eq. 2 here. See sect. 2.3 of Sánchez et al. (2020) for further details. Nevertheless, the residual between the full solution and the sum of both ($NO - HEAT + HEAT - ONLY$") is very small ( 1%). The residual of the case shown in the manuscript's Figure 13 is shown in Figure 2.d here. Given its marginal values we do not consider it is worth including it in Figure 13.

$$\nabla \cdot \mathbf{Q^{-1}}\nabla\left(\frac{\partial\phi}{\partial t}\right) = \nabla \cdot \mathbf{Q^{-1}H} + \nabla \cdot \vec{u}_g \tag{2}$$

[Figure]

Figure 2: As Figure 13 in the manuscript but including (d), the residual for $w$ (colour) and $\mathbf{v}_{ag}$ (vector). The residual is the difference between the "total" solution (panel a) minus the sum of no heat (panel b) plus the diabatic only (panel c) solutions.

L155: What is your estimate of the Rossby radius of deformation? 0.32° deg would be a rather small estimate. Why do you choose this length scale?
We first filter the MetUM fields to scales equivalent to the global operational models of 0.32°, as employed by Sánchez et al. (2020) and Hardy et al. (2023) where the SGT inversion tool was successfully applied. Then a second step is performed where fields are bi-linearly regridded to a grid resolution of 1.5° × 1.5°. This procedure allows reasonable computational accuracy in computing scales of 500km or greater (Cullen, 2018). The text is adapted accordingly at L176, copied below.
We filter scales below the Rossby radius of deformation in two steps. In the first step the discrete cosine filtering technique of Denis et al. (2002) is applied to filter wave-numbers above 50, equivalent to a wavelength of 0.32° and thus equivalent to the resolution of the global model employed in Sánchez et al. (2020) and Hardy et al. (2023), where the SGT inversion tool was successfully applied. In the second step the fields are bi-linearly regridded to a 1.5° × 1.5° latitude-longitude mesh so the SGT inversion tool can accurately solve the semi-geostrophic flow at and above the Rossby radius of deformation (Cullen, 2018) at a lower computational cost.

L208: This statement is not strictly correct because the relation is not monotonic. Please revise.
The text has been rewritten at L239, copied below
The maximum intensity of Ianos is higher in the +2 K SST simulations initialised at the three earliest times than in the latest two

L213: "trajectory" (here and elsewhere): "track" is the common term used for tropical-like storms. I suggest revising to avoid confusion.
Trajectory has been replaced by track at L218, caption of Fig. 1, L226, L245, caption of Fig. 3, L262 and L270.

L315: Titles of subsections 4.3 and 4.4, these subsections discuss also the tropospheric-deep PV tower not only the upper-level structure.
The section's title has been rewritten at L319, copied below
Evolution of the upper-level structure and associated PV-tower in the IFS analyses

L302: I have a hard time to identify the low PV values aloft. Please clarify or change color bar.
Figure 9 is updated to include blue contours for 0 PVU as shown in Figure 3 here, so it is now clearer where the blue areas (showing negative PV) are.

[Figure]

Figure 3: As manuscript's Fig 8 but including blue contours for $PV = 0$ $PUV$

L405: suggest adding: "size of the" dots.
Done at L460

**Anonymous Referee #2**

**1 General comments**

1. The tittle does not totally represent the content of the study and the conclusions: It focus more on the mechanisms that favor the development of Ianos rather than on the interaction between a warm sea or warmer sea and the convective factors that provoke Ianos cyclogenesis. It deals with this issue, but it is not the main part of the study, in my opinion. Indeed, in the abstract we do not see any focus on this.
See new title below, and the reply to second point in the major comments section of Reviewer #1.
The impact of preceding convection on the development of Medicane Ianos and the sensitivity to sea surface temperature

2. The study argues about the importance of baroclinic forcing in the development of Ianos, i.e. from a midlatitude meteorology point of view, but given it is a tropical transition phenomenon, it lacks addressing its development for a tropical meteorology point of view, i.e., the dynamics of convection and its organization. Therefore, I would explicitly mention this concern about the partial view of this phenomenon in this study when discussing the results and conclusions.
The new version of the manuscript includes several changes to relate our results to those from the tropical transition literature, see reply of the general point of reviewer #1.

**2 Specific points**

1. I would try to reduce the length of the article, especially Section 4.2, 4.3, 4.4 (which are not the main body of the line of arguments; or also merging 4.3 and 4.4 after summarising) and Conclusions and just focus on the main important aspects for understanding the main message about the mechanisms involved in Ianos' cyclogenesis and the role of the SST (although this is not mainly addressed).
   The importance of the results in former sections 4.2, 4.3 and 4.4 should be clearer in the new version of the manuscript. The split between former sections 4.3 and 4.4 is important, as 4.3 attributes the diabatic processes during the preceding process to the formation and maintenance of the low-PV bubble, and 4.4 the low-PV bubble to the baroclinicity triggering Ianos cyclogenesis.

2. Section 1 (Introduction):

   (a) L35-40: I would argue that upper-level baroclinic forcing is essential for the development of precursors to medicanes, not for the intensification of medicanes, at least for those robust (tropical) ones.
   The intensification of medicanes is not well understood and could be case dependent. The paragraphs containing these lines and the following paragraph have been rewritten at L38-L61, copied below.

   The upper-level cold air is provided by the intrusion of upper-level disturbances such as potential vorticity (PV) streamers or cut-off lows. The cyclogenesis of Medicanes thus occur under such moderate to strong baroclinic environments (Mazza et al., 2017; Fita and Flaounas, 2018; Flaounas et al., 2022). For instance, the reduction of the PV streamer by a PV inversion technique suppressed the cyclogenesis of a medicane in the cases described in Homar et al. (2003) and Carrió et al. (2017). Latent heating due to convection is also an important player in the intensification of Medicanes. Fita and Flaounas (2018) quantified the development of the surface pressure tendency for a medicane in December 2005 and attributed its deepening to warming in the atmospheric column from diabatic heating and advection within the upper troposphere. Convection is often fuelled by air-sea fluxes from the sea surface, e.g. the ensemble simulations of Noyelle et al. (2019), with different Sea Surface Temperature (SST) anomalies, relate the warm core strength to a linear increase of SST warming, but they also reveal that growth in the minimum pressure depth at maximum intensity is non-linear with the SST increase. Miglietta et al. (2011) found that increasing SST leads to a deeper medicane, stronger surface wind speeds and longer life-time of the medicane's tropical features.

   The relative importance of baroclinic instability and diabatic heating in medicane intensification seems to be case dependent. Some medicanes may not undergo a full tropical transition (Davis and Bosart, 2003), as suggested by Fita and Flaounas (2018) where their case classified as a subtropical cyclone. The climatology of tropical transitions of McTaggart-Cowan et al. (2013) shows high frequency of occurrence of strong tropical transition over the Mediterranean Sea, a region with a moderate to high coupling index McTaggart-Cowan et al. (2015). Miglietta and Rotunno (2019) proposed a classification of medicanes into three categories: (A) those where their later stages are dominated by the Wind-Induced Surface Heat Exchange mechanism (WISHE Emanuel, 1986; Rotunno and Emanuel, 1987), (B) those where the baroclinic instability is also important at later stages, and (C) a blend of the previous two where tropical transition and a dramatic intensification occurs after a short but intense interaction of the cyclone with an upper-level PV streamer. Miglietta and Rotunno (2019) conclude that the presence of a symmetric deep warm core does not imply full tropical dynamics and hence the terms "tropical-like" transition or "Mediterranean tropical-like cyclone" are often employed in the medicane literature.

3. Sections 4.1 and 4.2:

   (a) L197: What is the argument to say that it becomes a medicane at this time?
   It is stated on the previous section of the manuscript at L209, now referenced at L229). The new version has further details on the timings for medicane transition. See reply to general comment of reviewer # 1.

   (b) Figure 2: As the different initialization times are indicated by a different symbol, I would put just the same color for the CTL, -2K and +2K groups. In this way, the interpretation becomes clearer without too much noise.
   We would prefer the current colour system as it highlights how the simulations with similar

SST forcing change across initialisation times more clearly. Moreover, if we kept the same colours and symbols for all figures showing timeseries, which simplifies the reading, timeseries in Figure 5 would be even harder to differentiate if all would have the same colour.

(c) L215-225: It would be nice to make this analysis more complete by adding the CPS or at least the VTU values for each run, to provide a clearer idea of its tropical (or not) structure.
There is a new section "Ianos's tropical transition" at L299. It Includes the CPS diagnostic of Hart (2003) and the Brightness temperature (formerly Figure 4, now figure 7). The CPS computation is described at L107 and below
The tropical transition in Ianos is explored with the cyclone phase space (CPS) diagnostic of Hart (2003), with a couple of differences in its methodolody. First, our domain is small so we use a radius of 400 km instead of 500 km, see section 3.c of Mazza et al. (2017) for a discussion on the choice of different radius. Second, the thickness asymmetry term $B$ is computed using the 925 hPa level instead of 900 hPa, as the latter level is not available in our simulations.

(d) L233: Could you provide more evidence for the dry-eyes mechanism of triggering these thunderstorms.
The SEVIRI water vapour image omitted from the manuscript is shown in Fig. 4 here. There is a clear dry filament extending from a low-pressure system situated out in the southern North Atlantic, North of Madeira, crossing Northwesterly through the Northeast section of the Iberian Peninsula, passing south of the Balearic islands, bending above North Africa and ending on the Sidra Gulf where convection is taking place South of Sicily. This image is publicly accessible (after registration) from the referenced CEDA data centre.

[Figure]

Figure 4: SEVIRI water vapour image at $12Z$ 13, see manuscript for further details.

Further work could be done to attribute the event to the dry-eye mechanism, e.g. check whether the filament is collocated with a stratospheric PV filament. However, this attribution exercise is described with a speculative tone in our manuscript ("one possible mechanism ..."). We chose to avoid showing Fig. 4 and undertaking the attribution work to reduce the length of the paper. Detailing this process is beyond the scope of the paper.

4. Section 4.3:

(a) L304-305: Why the authors compare this development with those of extratropical cyclones and not with those of tropical transition? I would argue that this is a typical development for tropical transition cyclones and the result of Ianos acquiring a more robust tropical structure.
Panel (c) in Fig. 8 shows the $12Z$ 16 validation time, when Ianos has not yet developed an axisymmetric warm core (see new Figure 6). Panel (d) shows Ianos where it has already

developed the axisymmetric warm core. We have rewritten these lines at L356 to clarify this point, copied below, we also include a comparison with the composites of Galarneau et al. (2015).

At 12*Z* 17 Ianos completes its transition to a medicane (Fig. **??**): the positive PV anomaly in the low and mid-levels connects with the upper-level stratospheric intrusion (Fig. 3.d). PV towers are also a feature of tropical transitions of cyclones. For example, the composite of cyclones undergoing tropical transition of Galarneau et al. (2015) shows a better-defined PV tower extending to higher levels in the cases developing tropical transition than the non-developing cases.

(b) L307: Talking about tropical-like transition here is not suitable as this is not a rubust definition in the literature. The best way to deal with this type of development is talking about the tropical transition process, to be more in line with the community that study cyclone transitions.

The term "tropical-like transition" is defined at L59, copied below. We replace it by, or complement it with, axisymmetric warm core or medicane in L2, L288, L340, L356 and L555

Miglietta and Rotunno (2019) conclude that the presence of a symmetric deep warm core does not imply full tropical dynamics and hence the terms "tropical-like" transition or "Mediterranean tropical-like cyclone" are often employed in the medicane literature.

(c) In this section and the rest of the sections, I miss discussions about the upper-level dynamics that considers the jet streak behaviour. It is true that the PV field is related to the jet dynamics, but it would be interesting to also consider the analysis/explanations from the jet's QG forcing perspective in the analysis of this section and the following ones.

There is not a clear relationship between the position of the surface low and upper-level jets at the time of Ianos's cyclogenesis (Fig. 5.b). Ianos sits at the southern tip of the southerly flow while it intensifies one day later (Fig 5.c) as well as near the left jet exit region of the southwesterly jet streak. Hence, Ianos is in an optimal location for intensification relative to the southwesterly jet streak, but the interpretation considering both nearby jets is not straightforward.

[Figure]

Figure 5: Same as Figure 7 in the manuscript but showing wind speed contours instead of wind vectors, every 10 m/s.

5. Sections 4.5 and 4.6:

(a) MetUM means full solution?
Yes, regridded and filtered as detailed in the caption of Figure 12.

(b) L403: Is not the y-axis showing the dates? (Figure 14).

The description of Figure 14 in text and caption has been changed see reply to "(Fig. 14)" minor comment from reviewer #1

(c) L467 By watching Figure 16 (a-b), one could argue that the forcing at lower levels is stronger in (a) and, therefore, conclusions derived from Figure 15 are subject to the level of choice (700 hPa). If we choose 800 hPa or 850 hPa, for instance, conclusions could be different. Indeed, the signal seems to decrease a lot at the 700 hPa level. Please, elaborate more on this. Why not choose 800 hPa (it would still avoid boundary layer effects)? This argument of similar low-level forcing appears to be quite weak.

It is the level chosen by Deveson et al. (2002). See reply to the first major comment of reviewer #1 ("e" in particular). The section has been entirely rewritten at L506, with new figures.

6. Section 6 (conclusions):

(a) L538-544: I see the idea of the authors here, but I don't fully see the relationship between Ianos' development and this mechanism in extratropical transitions and downstream cascade of events. The link is about the upper-level PV modification by a diabatic source, but just up to this point.

Although we are considering a medicane rather than a tropical cyclone here, the extratropical transition of a TC is relevant as it is an example of diabatic processes preconditioning environmental baroclinicity and cyclone reinforcement. Thus we think it is relevant to highlight this evolution for those readers interested in how diabatic processes and baroclinicity interact with each other. The paragraph has been rewritten at L608 to make this point clearer, see below

The role of preceding or preconditioning precipitation events has been identified here during medicane development for the first time. There is another kind of cyclone transition where preceding precipitation also preconditions the baroclinic environment: the extra-tropical transition of tropical cyclones. During such events, tropical cyclones advect warm and moist air poleward, triggering extratropical cyclogenesis and initiating ridge building and jet acceleration ahead of the tropical cyclone (Grams and Archambault, 2016), which contributes to the re-intensification of the transitioning (tropical to extra-tropical) cyclone (Keller et al., 2019). In the Ianos case, the preceding precipitation is independent of the developing medicane.

(b) L541-544: An analysis focusing on the tropical transition perspective would make the article more complete.

The paragraph has been replaced at L603, copied below. It details the TT pathway from meridional thought precursors from Bentley et al. (2017), with further references to TT processes.

Ianos's pathway into a medicane shares similarities with the meridional trough category of subtropical cyclones undergoing tropical transition (Bentley et al., 2017). The development of deep convection in this category may be partially induced by QG forcing; these cyclones show a weaker dependency on bulk tropospheric stability and a greater frequency of strong tropical transition events, those able to produce a WISHE mechanism under a pronounced weakening of shear environment from convection upshear (Davis and Bosart, 2003).

(c) 545-552 I think this part is not very well connected with Ianos' mechanisms and it could derive some confusion.

These lines have been removed and replaced by the two paragraphs detailed in the two replies above, at L603-613 in the new version of the manuscript.

(d) L562: I would not mention this hypothetical situation as (as you discuss later) the model is uncoupled and a robust analysis about model performance in this case has not been undergone. This scenario could be quite fictitious.

We think this paragraph provides enough context to understand how "fictitious" our simulations with +2K SST could be. Its purpose is mainly to motivate future work to understand the role of SST under a changing climate and its possible impact on medicane development, as well as to advocate for the use of regional coupled models at convective scales for climate studies. The end of the last sentence of the paragraph has been rewritten at L626, copied below. It now re-focuses into a modelling perspective, where our results are more relevant.

An accurate representation of SST changes and air-sea interactions in convective-scale regional climate prediction models may be critical to determining the future climatology of medicanes.

**References**

A. M. Bentley, D. Keyser, and L. F. Bosart. A dynamically based climatology of subtropical cyclones that undergo tropical transition in the north atlantic basin. *Monthly Weather Review*, 144(5):2049 – 2068, 2016. doi: 10.1175/MWR-D-15-0251.1.

A. M. Bentley, L. F. Bosart, and D. Keyser. Upper-tropospheric precursors to the formation of subtropical cyclones that undergo tropical transition in the north atlantic basin. *Monthly Weather Review*, 145(2): 503 – 520, 2017. doi: 10.1175/MWR-D-16-0263.1.

M. Bush, I. Boutle, J. Edwards, A. Finnenkoetter, C. Franklin, K. Hanley, A. Jayakumar, H. Lewis, A. Lock, M. Mittermaier, S. Mohandas, R. North, A. Porson, B. Roux, S. Webster, and M. Weeks. The second met office unified model–jules regional atmosphere and land configuration, ral2. *Geosci. Model Dev.*, 16(6):1713–1734, 2023. doi: 10.5194/gmd-16-1713-2023.

M. Bush, D. L. Flack, A. Arnold, M. Best, S. I. Bohnenstengel, I. Boutle, J. Brooke, S. Cole, S. Cooper, G. Dow, J. Edwards, P. Field, A. Finnenkoetter, C. Franklin, K. Furtado, K. Halladay, K. Hanley, M. Hendry, A. Hill, A. Jayakumar, R. Jones, H. Lewis, J. Lee, A. Lock, A. McCabe, M. Mittermaier, S. Mohandas, S. Moore, C. Morcrette, R. North, A. Porson, S. Rennie, B. Roux, C. Sanchez, C. Short, C.-H. Su, S. Tucker, K. V. Weverberg, S. Vosper, D. Walters, J. Warner, S. Webster, M. Weeks, J. Wilkinson, M. Whitall, K. Williams, and H. Zhang. Unifying mid-latitude and tropical regional model configurations: The third met office unified model–jules regional atmosphere and land configuration, ral3. *Geosci. Model Dev.*, 2024. in preparation.

D. Carrió, V. Homar, A. Jansa, R. Romero, and M. Picornell. Tropicalization process of the 7 november 2014 mediterranean cyclone: Numerical sensitivity study. *Atmos. Res.*, 197:300–312, 2017. ISSN 0169-8095. doi: 10.1016/j.atmosres.2017.07.018.

M. Cullen. The use of semigeostrophic theory to diagnose the behaviour of an atmospheric gcm. *Fluids*, 3(4), 2018. ISSN 2311-5521. doi: 10.3390/fluids3040072.

C. A. Davis and L. F. Bosart. Baroclinically induced tropical cyclogenesis. *mwr*, 131(11):2730 – 2747, 2003. doi: 10.1175/1520-0493(2003)131¡2730:BITC¿2.0.CO;2.

B. Denis, J. Côté, and R. Laprise. Spectral decomposition of two-dimensional atmospheric fields on limited-area domains using the discrete cosine transform (dct). *mwr*, 130(7):1812 – 1829, 2002.

A. C. L. Deveson, K. A. Browning, and T. D. Hewson. A classification of FASTEX cyclones using a height-attributable quasi-geostrophic vertical-motion diagnostic. *QJRMS*, 128(579):93–117, Jan. 2002. doi: 10.1256/00359000260498806.

K. A. Emanuel. An air-sea interaction theory for tropical cyclones. part i: Steady-state maintenance. *jas*, 43(6):585 – 605, 1986. doi: 10.1175/1520-0469(1986)043¡0585:AASITF¿2.0.CO;2.

J. L. Evans and M. P. Guishard. Atlantic subtropical storms. part i: Diagnostic criteria and composite analysis. *Monthly Weather Review*, 137(7):2065 – 2080, 2009. doi: 10.1175/2009MWR2468.1.

P. R. Field, A. J. Heymsfield, and A. Bansemer. Snow size distribution parameterization for midlatitude and tropical ice clouds. *Journal of the Atmospheric Sciences*, 64(12):4346 – 4365, 2007. doi: 10.1175/2007JAS2344.1.

P. R. Field, A. Hill, B. Shipway, K. Furtado, J. Wilkinson, A. Miltenberger, H. Gordon, D. P. Grosvenor, R. Stevens, and K. Van Weverberg. Implementation of a double moment cloud microphysics scheme in the uk met office regional numerical weather prediction model. *Quarterly Journal of the Royal Meteorological Society*, 149(752):703–739, 2023. doi: https://doi.org/10.1002/qj.4414.

L. Fita and E. Flaounas. Medicanes as subtropical cyclones: the december 2005 case from the perspective of surface pressure tendency diagnostics and atmospheric water budget. *qjrms*, 144(713):1028–1044, 2018. doi: 10.1002/qj.3273.

E. Flaounas, S. Davolio, S. Raveh-Rubin, F. Pantillon, M. M. Miglietta, M. A. Gaertner, M. Hatzaki, V. Homar, S. Khodayar, G. Korres, V. Kotroni, J. Kushta, M. Reale, and D. Ricard. Mediterranean cyclones: current knowledge and open questions on dynamics, prediction, climatology and impacts. *Weather Clim. Dynam.*, 3(1):173–208, 2022. doi: 10.5194/wcd-3-173-2022.

T. J. Galarneau, R. McTaggart-Cowan, L. F. Bosart, and C. A. Davis. Development of north atlantic tropical disturbances near upper-level potential vorticity streamers. *Journal of the Atmospheric Sciences*, 72(2):572 – 597, 2015. doi: 10.1175/JAS-D-14-0106.1.

C. M. Grams and H. M. Archambault. The key role of diabatic outflow in amplifying the midlatitude flow: A representative case study of weather systems surrounding western north pacific extratropical transition. *mwr*, 144(10):3847 – 3869, 2016. doi: 10.1175/MWR-D-15-0419.1.

S. L. Gray and H. F. Dacre. Classifying dynamical forcing mechanisms using a climatology of extratropical cyclones. *qjrms*, 132(617):1119–1137, 2006. doi: https://doi.org/10.1256/qj.05.69.

S. Hardy, J. Methven, J. Schwendike, B. Harvey, and M. Cullen. Examining the dynamics of a borneo vortex using a balance approximation tool. *EGUsphere*, 2023:1–31, 2023. doi: 10.5194/egusphere-2023-1312.

R. E. Hart. A cyclone phase space derived from thermal wind and thermal asymmetry. *mwr*, 131(4):585 – 616, 2003. doi: 10.1175/1520-0493(2003)131¡0585:ACPSDF¿2.0.CO;2.

V. Homar, R. Romero, D. J. Stensrud, C. Ramis, and S. Alonso. Numerical diagnosis of a small, quasi-tropical cyclone over the western mediterranean: Dynamical vs. boundary factors. *qjrms*, 129(590):1469–1490, 2003. doi: 10.1256/qj.01.91.

B. J. Hoskins, I. Draghici, and H. C. Davies. A new look at the omeaga equation. *Quarterly Journal of the Royal Meteorological Society*, 104(439):31–38, 1978. doi: https://doi.org/10.1002/qj.49710443903.

J. H. Keller, C. M. Grams, M. Riemer, H. M. Archambault, L. Bosart, J. D. Doyle, J. L. Evans, T. J. Galarneau, K. Griffin, P. A. Harr, N. Kitabatake, R. McTaggart-Cowan, F. Pantillon, J. F. Quinting, C. A. Reynolds, E. A. Ritchie, R. D. Torn, and F. Zhang. The extratropical transition of tropical cyclones. part ii: Interaction with the midlatitude flow, downstream impacts, and implications for predictability. *mwr*, 147(4):1077 – 1106, 2019. doi: 10.1175/MWR-D-17-0329.1.

K. Lagouvardos, A. Karagiannidis, S. Dafis, A. Kalimeris, and V. Kotroni. Ianos—a hurricane in the mediterranean. *BAMS*, 103(6):E1621 – E1636, 2022. doi: 10.1175/BAMS-D-20-0274.1.

E. Mazza, U. Ulbrich, and R. Klein. The tropical transition of the october 1996 medicane in the western mediterranean sea: A warm seclusion event. *mwr*, 145(7):2575 – 2595, 2017. doi: 10.1175/MWR-D-16-0474.1.

R. McTaggart-Cowan, G. D. Deane, L. F. Bosart, C. A. Davis, and T. J. Galarneau. Climatology of tropical cyclogenesis in the north atlantic (1948–2004). *Monthly Weather Review*, 136(4):1284 – 1304, 2008. doi: 10.1175/2007MWR2245.1.

R. McTaggart-Cowan, T. J. Galarneau, L. F. Bosart, R. W. Moore, and O. Martius. A global climatology of baroclinically influenced tropical cyclogenesis. *Monthly Weather Review*, 141(6):1963 – 1989, 2013. doi: 10.1175/MWR-D-12-00186.1.

R. McTaggart-Cowan, E. L. Davies, J. G. Fairman, T. J. Galarneau, and D. M. Schultz. Revisiting the 26.5°c sea surface temperature threshold for tropical cyclone development. *Bulletin of the American Meteorological Society*, 96(11):1929 – 1943, 2015. doi: 10.1175/BAMS-D-13-00254.1.

M. M. Miglietta and R. Rotunno. Development mechanisms for mediterranean tropical-like cyclones (medicanes). *qjrms*, 145(721):1444–1460, 2019. doi: 10.1002/qj.3503.

M. M. Miglietta, A. Moscatello, D. Conte, G. Mannarini, G. Lacorata, and R. Rotunno. Numerical analysis of a mediterranean 'hurricane' over south-eastern italy: Sensitivity experiments to sea surface temperature. *Atmos. Res.*, 101(1):412–426, 2011. ISSN 0169-8095. doi: 10.1016/j.atmosres.2011.04.006.

M. M. Miglietta, S. Laviola, A. Malvaldi, D. Conte, V. Levizzani, and C. Price. Analysis of tropical-like cyclones over the mediterranean sea through a combined modeling and satellite approach. *grl*, 40(10):2400–2405, 2013. doi: 10.1002/grl.50432.

M. M. Miglietta, D. Cerrai, S. Laviola, E. Cattani, and V. Levizzani. Potential vorticity patterns in mediterranean "hurricanes". *grl*, 44(5):2537–2545, 2017. doi: 10.1002/2017GL072670.

M. C. Morgan. Using piecewise potential vorticity inversion to diagnose frontogenesis. part i: A partitioning of the q vector applied to diagnosing surface frontogenesis and vertical motion. *Monthly Weather Review*, 127(12):2796 – 2821, 1999. doi: 10.1175/1520-0493(1999)127¡2796:UPPVIT¿2.0.CO;2.

R. Noyelle, U. Ulbrich, N. Becker, and E. P. Meredith. Assessing the impact of sea surface temperatures on a simulated medicane using ensemble simulations. *Nat. Hazards Earth Syst. Sci.*, 19(4):941–955, 2019. doi: 10.5194/nhess-19-941-2019.

F. Pantillon, S. Davolio, E. Avolio, C. Calvo-Sancho, D. S. Carrió, S. Dafis, E. Flaounas, E. S. Gentile, J. J. Gonzalez-Aleman, S. Gray, M. M. Miglietta, P. Patlakas, I. Pytharoulis, D. Ricard, A. Ricchi, and C. Sanchez. The crucial representation of deep convection for the cyclogenesis of medicane ianos. *EGUsphere*, 2024:1–29, 2024. doi: 10.5194/egusphere-2024-1105.

R. S. Plant, G. C. Craig, and S. L. Gray. On a threefold classification of extratropical cyclogenesis. *qjrms*, 129(594):2989–3012, 2003. doi: https://doi.org/10.1256/qj.02.174.

R. Rotunno and K. A. Emanuel. An air–sea interaction theory for tropical cyclones. part ii: Evolutionary study using a nonhydrostatic axisymmetric numerical model. *jas*, 44(3):542 – 561, 1987. doi: 10.1175/1520-0469(1987)044¡0542:AAITFT¿2.0.CO;2.

C. Sánchez, J. Methven, S. Gray, and M. Cullen. Linking rapid forecast error growth to diabatic processes. *qjrms*, 146(732):3548–3569, 2020. doi: 10.1002/qj.3861.

B. Thompson, C. Sanchez, X. Sun, G. Song, J. Liu, X.-Y. Huang, and P. Tkalich. A high-resolution atmosphere–ocean coupled model for the western maritime continent: development and preliminary assessment. *Climate Dynamics*, 52:3951–3981, 2019. doi: 10.1007/s00382-018-4367-0.

M. Tous and R. Romero. Meteorological environments associated with medicane development. *International Journal of Climatology*, 33(1):1–14, 2013. doi: https://doi.org/10.1002/joc.3428.

H. Wernli and H. C. Davies. A lagrangian-based analysis of extratropical cyclones. i: The method and some applications. *Quarterly Journal of the Royal Meteorological Society*, 123(538):467–489, 1997. doi: https://doi.org/10.1002/qj.49712353811. URL https://rmets.onlinelibrary.wiley.com/doi/abs/10.1002/qj.49712353811.

W. Yanase, U. Shimada, N. Kitabatake, and E. Tochimoto. Tropical transition of tropical storm kirogi (2012) over the western north pacific: Synoptic analysis and mesoscale simulation. *Monthly Weather Review*, 151(10):2549 – 2572, 2023. doi: 10.1175/MWR-D-22-0190.1.

F. Zimbo, D. Ingemi, and G. Guidi. The tropical-like cyclone "ianos" in september 2020. *Meteorology*, 1 (1):29–44, 2022. doi: 10.3390/meteorology1010004.

---

## Author Response (AR2)

**point-by-point response to $2^{nd}$ rev. of *egusphere-2023-2431**

We keep the same format as the previous "point-by-point" review, the reply to reviewers comments is written in blue here, and changes to the manuscript in red. Line numbers in my replies refer to the newer revision where the comments here are addressed.

**Anonymous Referee #1**

- L176ff: Why is the filtering needed before regridding? Does the regridding involve averaging of 0.32 deg information? The procedure is unclear to me and suggests that the 1.5 grid still contains local extremes from the 0.32 data, which would require justification why such extremes were representative. Why not done similarly as for the QG Omega Eq below? Please clarify.

  We acknowledge that the filtering methodology for the SGT inversion tool may lead to confusion. However, the tool has barely been employed in convection-permitting resolutions and limited area models, ours would be the first publication to report its results. We were thus concerned about the computational cost and numerical stability of the tool with inputs at such scales. Hence we decided to apply the spectral regridding to a 0.32 deg resolution, in order to reduce the extremes of unbalanced dynamics in and around convective cores, the 0.32 deg resolution is slightly coarser than the global models with parametrized convection employed by Sánchez et al. (2020) and Hardy et al. (2023). Afterwards, we applied the 1.5 deg regridding step done in both publications.

  Filtering to a lower wavenumber, equivalent to a 1.5km resolution, would retain the same grid-spacing. We are not sure how well the SGT inversion code could computationally cope with the larger arrays, and how it would converge at such very high resolutions. On the other hand, regridding to 1.5km without the spectral smoothing migh not be effective enough to reduce the unbalanced high vertical winds from the convection permitting scales, and again, we are not sure about how it will impact the convergence to the solution of the SGT inversion. The relevant lines are rewritten in L172, copied below

  Due to concerns about how the SGT inversion tool will converge with "noisy" vertical motions in the 2.2 km convective-scale model output (as previous use of this tool has used convection-parametrizing global model output), we first filter scales above wave-number 50, equivalent to a wavelength of 0.32°, following the discrete cosine filtering technique of Denis et al. (2002). High vertical winds in and around convective cores are thus filtered and are equivalent to the ones in the global model employed by Sánchez et al. (2020) and Hardy et al. (2023), where the SGT inversion tool was successfully applied. Then we follow their methodology to bi-linearly regrid the fields to a $1.5° \times 1.5°$ ...

- L301: Why "even"? Increasing vertical shear seems to be consistent with a non-developing system. I suggest revising.

  The Sentence is rewritten at L301 and copied below, to show there are negligible differences in wind shear amongst simulations developing or failing to develop medicane Ianos.

  with negligible differences between simulations developing and not developing medicane Ianos

- L302: What is meant by "neutral"? Please clarify?

  Replaced by "weak" in L303

- L306ff: Does the increase of the thermal wind imply that the warm core strengthens? Please clarify for the non-experts of the phase-space diagrams.

  Added the sentence below at L307

  which implies that the warm core strengthens

- L312: What is meant here with "branches"? Do you refer to rain/ cloud bands? I suggest revising.

  Replaced by "cloud bands" at L313

- L315: Extending to a radius or a diameter of 500km? Please clarify.

  Extending to a diameter of approx 500km. Corrected in the text at L316

- L326: What is meant with "blending" in this context? I suggest revising.

  Replaced by "mixing" at L327.

- Pg17, discussion of PV tower: It should be noted that such a tropospheric-deep, positive PV anomaly is indeed a key feature of tropical cyclones, too.

  The following sentence is added at L361

  Such a troposphere-deep, positive PV anomaly is also a key feature of tropical cyclones (Thorpe, 1985).

- L363: How do I see the streamer intruding into the troposphere? What level is meant with lower levels? The PV tower has a warm core (high theta) up to 300 hPa, indicating diabatic origin.

  The sentence is rewritten at L365, copied below

  The PV streamer south of the PV tower intrudes into the troposphere to levels below 300 hPa.

- L509: How do we see the coupling here? We see different evolutions but how do you diagnose coupling from this figure?

  "coupling" is replaced by "connection" at L511. These couple of sentences aim to make the point that the three fields (surface pressure core, low-PV bubble and diabatic divergence aloft) look more aligned or connected in Fig. 15.h, the "good forecast", than in Fig 15.e the "poor forecast".

**Anonymous Referee #2**

- **Previous comment on Sections 4.1 and 4.2:(a) L197:** What is the argument to say that it becomes a medicane at this time? It is stated on the previous section of the manuscript at L209, now referenced at L229).

  *Previous reply to comment:* The new version has further details on the timings for medicane transition. See reply to general comment of reviewer #1.

  *New comment from reviewer:* As we do not have a clear definition of "Medicane" I would just say "when Ianos became a deep warm core".

  "Medicane" is replaced by "develops a deep warm core" at L230.

- **Previous comment on Section 4.3: (b) L307:** Talking about tropical-like transition here is not suitable as this is not a rubust definition in the literature. The best way to deal with this type of development is talking about the tropical transition process, to be more in line with the community that study cyclone transitions.

  *Previous reply to comment:* The term "tropical-like transition" is defined at L59, copied below. We replace it by, or complement it with, axisymmetric warm core or medicane in L2, L288, L340, L356 and L555 Miglietta and Rotunno (2019) conclude that the presence of a symmetric deep warm core does not imply full tropical dynamics and hence the terms "tropical-like" transition or "Mediterranean tropical-like cyclone" are often employed in the medicane literature.

  *New comment from reviewer:* This is related to the previous comment.

  Done at L355. All other medicane entries in the Results section are to describe "Medicane Ianos" or to describe results from cited publications. We keep these unchanged to maintain traceability with the literature.

- **Previous comment on Sections 4.5 and 4.6:** (c) L467 By watching Figure 16 (a-b), one could argue that the forcing at lower levels is stronger in (a) and, therefore, conclusions derived from Figure 15 are subject to the level of choice (700 hPa). If we choose 800 hPa or 850 hPa, for instance, conclusions could be different. Indeed, the signal seems to decrease a lot at the 700 hPa level. Please, elaborate more on this. Why not choose 800 hPa (it would still avoid boundary layer effects)? This argument of similar low-level forcing appears to be quite weak.

*Previous reply to comment:* It is the level chosen by Deveson et al. (2002). See reply to the first major comment of reviewer #1 ("e" in particular). The section has been entirely rewritten at L506, with new figures.

*New comment from reviewer:* There is a mismatch in Figure 16 regarding times in the head of the figures and figure captions and about "good" and "bad" simulations. (a) is the good simulation and (b) is the bad simulation.

Yes, reference to panels and initialisation times in the caption is wrong, thanks for spotting it! Fig 16 caption is corrected in the new manuscript, and copied below.

(a) $00Z\,15$, defined as the "good simulation" in the text, and (b) $12Z\,14$, the "poor simulation"

**References**

M. Cullen. The use of semigeostrophic theory to diagnose the behaviour of an atmospheric gcm. *Fluids*, 3(4), 2018. ISSN 2311-5521. doi: 10.3390/fluids3040072.

M. J. P. Cullen, R. J. Douglas, I. Roulstone, and M. J. Sewell. Generalized semi-geostrophic theory on a sphere. *jfm*, 531:123–157, 2005. doi: 10.1017/S0022112005003812.

B. Denis, J. Côté, and R. Laprise. Spectral decomposition of two-dimensional atmospheric fields on limited-area domains using the discrete cosine transform (dct). *mwr*, 130(7):1812 – 1829, 2002. doi: 10.1175/1520-0493(2002)130¡1812:SDOTDA¿2.0.CO;2.

S. Hardy, J. Methven, J. Schwendike, B. Harvey, and M. Cullen. Examining the dynamics of a borneo vortex using a balance approximation tool. *EGUsphere*, 2023:1–31, 2023. doi: 10.5194/egusphere-2023-1312.

K. Lagouvardos, A. Karagiannidis, S. Dafis, A. Kalimeris, and V. Kotroni. Ianos—a hurricane in the mediterranean. *bams*, 103(6):E1621 – E1636, 2022. doi: 10.1175/BAMS-D-20-0274.1. URL `https://journals.ametsoc.org/view/journals/bams/103/6/BAMS-D-20-0274.1.xml`.

C. Sánchez, J. Methven, S. Gray, and M. Cullen. Linking rapid forecast error growth to diabatic processes. *qjrms*, 146(732):3548–3569, 2020. doi: 10.1002/qj.3861.

A. J. Thorpe. Diagnosis of balanced vortex structure using potential vorticity. *Journal of Atmospheric Sciences*, 42(4):397 – 406, 1985. doi: 10.1175/1520-0469(1985)042¡0397:DOBVSU¿2.0.CO;2.